# TEST-TIME ADAPTATION AND ADVERSARIAL ROBUSTNESS

## ABSTRACT

This paper studies test-time adaptation in the context of adversarial robustness. We formulate an adversarial threat model for test-time adaptation, where the defender may have a unique advantage as the adversarial game becomes a maximin game, instead of a minimax game as in the classic adversarial robustness threat model. We then study whether the maximin threat model admits more "good solutions" than the minimax threat model, and is thus *strictly weaker*. For this purpose, we first present a provable separation between the two threat models in a natural Gaussian data model. For deep learning, while we do not have a proof, we propose a candidate, Domain Adversarial Neural Networks (DANN), an algorithm designed for unsupervised domain adaptation, by showing that it provides nontrivial robustness in the test-time maximin threat model against strong transfer attacks and adaptive attacks. This is somewhat surprising since DANN is not designed specifically for adversarial robustness (e.g., against norm-based attacks), and provides no robustness in the minimax model. Complementing these results, we show that recent data-oblivious test-time adaptations can be easily attacked even with simple transfer attacks. We conclude the paper with various future directions of studying adversarially robust test-time adaptation.

## 1 INTRODUCTION

There is a surge of interest to study test-time adaptation to help generalization to unseen domains (e.g., recent work by Sun et al. (2020); Wang et al. (2020); Nado et al. (2020)). At the high level, a generic test-time adaptation can be modeled as an algorithm $\Gamma$ which accepts an (optional) labeled training dataset $D$, an (optional) model $F$ trained on $D$ (usually used as a starting point), and an unlabeled test feature set $U$, outputs a model $\widetilde{F} = \Gamma(F, D, U)$, in order to achieve high test accuracy on $U$. For large test set $U$, test-time adaptation can be viewed as a form of *transductive learning* (Joachims (1999); Vapnik (1998)) (i.e., using $D, U$ to train a model to predict specific instances in $U$), which is argued to be easier than more traditional inductive learning.

This paper studies test-time adaptation in the context of adversarial robustness (i.e., there is an active agent who tries to fool the test-time adaptation by perturbing the input so that $\widetilde{F}$ gives wrong predictions). There are several motivations in pursuing this direction. First, this question is of practical interest: Many practical ML pipelines run in a *batch mode*[1], where they first collect a set of unlabelled data points, and then send them to a model (e.g. Nado et al. (2020)). In such cases, data in the batch may have been adversarially perturbed, and it is a natural question *whether we can leverage the large batch size and test-time adaptation to enhance adversarial robustness.* Second, from a purely theoretical point of view, since test-time adaptation is a form of transductive learning, it is intriguing to ask *whether transductive adversarial learning can be easier*, given that traditional adversarial robustness is formulated in the *inductive* learning setting (e.g. Madry et al. (2018)). To this end, a recent work by Goldwasser et al. (2020) shows that, with transductive learning, one can achieve nontrivial guarantees for classes of bounded VC dimension with *arbitrary* train and test distributions. The current work complements their paper in the setting of deep learning.

To study these questions, we formalize a threat model, which we call *(test-time) maximin threat model*, for the adversarial robustness of test-time adaptation. Recall that the classic adversarial

---

[1]For example,Instagram collects a large batch of photos before sending them to a model to tag people.

robustness game is a minimax game $\min_F \mathbb{E}_V[\max_{\widetilde{V}} L(F, \widetilde{V})]$, where $V$ is random sampled data, $\widetilde{V}$ is the perturbed data generated from $V$ by the adversary, and $L(F, \widetilde{V})$ is the loss of the model $F$ on $\widetilde{V}$. By contrast, in the maximin threat model, we allow $V$ to be sampled from a different domain, and the game is maximin: $\mathbb{E}_V[\max_U \min_{\widetilde{F}} L(\widetilde{F}, \widetilde{V})]$ (where $U$ is the perturbed features of $V$, subject to the attack type, and $\widetilde{V}$ is the labeled perturbed data, see Definition 2). By the maximin inequality, it follows that this threat model is *no harder* than the minimax model (to allow source and target domains to be different, we need to generalize the classic minimax model, see Definition 3).

We then move on to the focus of this work: Whether the maximin threat model is "strictly weaker" than the minimax threat model. We note that any good defender solution (a robust model) in the minimax game induces a good defender solution in the maximin game (an adaptation algorithm that outputs that robust model), thus intuitively, the good defender solutions of the minimax model is a subset of the good defender solutions of the maximin threat model. We ask *whether such a containment is proper*: That is, whether there *exists* a defender solution that is good in the maximin threat model, but is bad in the minimax threat model. The existence of such a defender will demonstrate that the maximin threat model admits more good solutions. Besides theoretical interest, this question is also of practical importance since these "new" solutions may possess *desirable properties that good solutions in the minimax threat model may lack*. For example, one such property is that the defender solution is *attack agnostic* (Goodfellow (2018) (pp.30)): That is, the solution is not to directly optimize the performance measure for a particular type of perturbation[2].

To this end, we first present a *provable separation* between the maximin and minimax threat models in a natural Gaussian data model. In fact, the separation holds even when $U$ only contains *a single point*, indicating the power of transductive learning. We then move to deep learning. While we do not have provable guarantees, we empirically examine Domain Adverarial Neural Networks (DANN) (Ganin et al. (2017)), an algorithm designed for *unsupervised domain adaptation* (UDA), as a *candidate* for the separation. Specifically, we demonstrate that DANN provides nontrivial test-time adversarial robustness against both *transfer attacks* and *adaptive attacks*, in both homogeneous and inhomogeneous cases. This is somewhat surprising as DANN is attack agnostic as we mentioned above, and has not been considered for adversarial robustness. Not surprisingly, as we hypothesized for a separation, the accuracy becomes very low when evaluating $\widetilde{F}$ in the minimax model.

Complementing the above result, we explore the maximin robustness of the recent data-oblivious adaptation algorithms (namely, the adaptation algorithms do not use $D$, but just the pretrained model $F$ and unlabeled test set $U$). Specifically, we consider Test-Time Training (TTT) by Sun et al. (2020)[3]. We show that TTT can be easily attacked using simple transfer attacks. While this is not surprising as authors of Sun et al. (2020) have cautioned that TTT is not designed for adversarial robustness, the situation is in sharp contrast to our results with DANN.

The rest of the paper is organized as follows: Section 2 presents the setup. In Section 3 we define threat models. In Section 4 we present theoretical results about separation, and examine DANN as a candidate separation in the deep learning. Finally, Section 5 explores the maximin robustness of oblivious test-time adaptation, and concludes the paper with future directions.

## 2 PRELIMINARIES

Let $F$ be a model, for a data point $(x, y) \in \mathcal{X} \times \mathcal{Y}$, a loss function $\ell(F; x, y)$ give the loss of $F$ on $x$ given the true label $y$. Let $V$ be a set of labeled data points. We use the notation $L(F, V) = \frac{1}{|V|} \sum_{(x,y) \in V} \ell(F; x, y)$ to denote the empirical loss of $F$ on $V$. For example, if we use binary loss $\ell^{0,1}(F; x, y) = \mathbb{1}[F(x) \neq y]$, this gives the test error of $F$ on $V$. We use the notation $V|_X$ to denote the projection of $V$ to its features, that is $\{(x_i, y_i)\}_{i=1}^m \mapsto \{x_1, \ldots, x_m\}$.

**Threat model for classic adversarial robustness.** To formulate the threat model for test-time adaptation, we first present a threat model for the classic adversarial robustness. Although the classic adversarial robustness can be written down succinctly as a minimax objective, namely

---

[2]Another consideration, which is beyond the scope of this paper, is the computational feasibility of finding a good solution, given the hardness of minimax optimization Katz et al. (2017); Daskalakis et al. (2020).

[3]While TTT does not use training data $D$ at the test time, it has a special self-training component, and the joint architecture is a $Y$-structure. A more domain agnostic approach is discussed in Wang et al. (2020).

$\min_F \mathbb{E}_{(x,y)\sim(X,Y)} \left[ \max_{x'\in N(x)}[\ell(F; x', y)] \right]$ ($N(x)$ is a neighborhood function of $x$, determined by the attack type), a threat model formulation will help us develop more nuanced models.

---

**Definition 1** (**Threat model for classic adversarial robustness**). *Attacker and defender agree on a particular attack type. Attacker is an algorithm $\mathcal{A}$, and defender is a supervised learning algorithm $\mathcal{T}$.*

**Before game starts**
- *A (labeled) training set $D$ is sampled i.i.d. from from $(X, Y)$.*

**Training time**
- *(**Defender**) Train a model $F$ on $D$ as $F = \mathcal{T}(D)$.*

**Test time**
- *A (labeled) natural test set $V$ is sampled i.i.d. from $(X, Y)$.*
- *(**Attacker**) On input $F$, $D$, and $V$, $\mathcal{A}$ perturbs each point $(x, y) \in V$ to $(x', y)$ (subject to the agreed attack type), giving $\widetilde{V} = \mathcal{A}(F, D, V)$.*

**Evaluation:**
*Evaluate the test loss of $F$ on $\widetilde{V}$, $L(F, \widetilde{V})$. Attacker's goal is to maximize the test loss, while the defender's goal is to minimize the test loss.*

---

We stress that the i.i.d sampling of $V$ is important (which is also present in the expectation in the minimax objective): Otherwise an attacker can pick any point that fools $F$ and repeat it arbitrarily many times. (we refer readers to Goodfellow (2019) for more discussions along this line).

**Notations for models and attacks.** In this paper we mainly use the PGD attacks (Projected Gradient Descent attacks) with norm-based perturbations Madry et al. (2018). For example, given a model $F$, we use the notation $\text{PGD}(F, V)$ to denote PGD attacks against $F$, on data $V$ (the attack type is specified in the context). We adopt the following notations:

| Notation | Meaning |
|---|---|
| T | A target model trained on the labeled target data $V$. |
| AdvT | An adversarially trained target model using the labeled target data $V$. |
| S | A source model trained on the labeled source data $D$. |
| AdvS | An adversarially trained source model using the labeled source data $D$. |
| $\text{PGD}(\cdot, \cdot)$ | PGD Attacks on a model and data. For example, $\text{PGD}(\text{AdvT}, V)$ means to use PGD attacks on the model AdvT and data $V$. |

**Test-time defenses and BPDA.** Various previous work have investigated *test-time defenses* where a pretrained model is fixed and there is a "preprocessing procedure" to preprocess an input before sending it to the model. Several such defenses were described and attacked in Athalye et al. (2018), by the BPDA technique (Backward Pass Differentiable Approximation). While syntactically one can fit these defenses into our framework, they only form some very special cases of our framework, which reuses a fixed pretrained model and focuses on *input sanitization*. As we will show later in the paper, for both of our provable separation and deep learning results, the adaptation algorithms *train new models* (beyond sanitizing inputs); and theoretically attacking these adaptations becomes a bilevel optimization. In these cases, it is unclear how to apply BPDA, and indeed it is an intriguing direction to further study attacking unsupervised domain adaptation algorithms, such as DANN.

## 3 THREAT MODELS

### 3.1 TEST-TIME MAXIMIN THREAT MODEL

The intuition behind the test-time maximin threat model is as follows: After we receive the adversarially perturbed data $U$ to classify (at test time), the defender trains a model based on $U$, and we evaluate the test accuracy only for $U$. (i.e., for different test set $U$ we may have different models and different test accuracy.) This perspective of training a model using labeled data $D$ and unlabeled data $U$, and only test on $U$, is essentially the *transductive learning* by Vapnik (1998) (however, we consider it in an adversarial setting). Formally, we have the following definition:

---

**Definition 2** (**Maximin threat model for test-time adaptation**). *Fix an adversarial perturbation type. We have a source domain $(X^s, Y^s)$, and a target domain $(X^t, Y^t)$. Attacker is an algorithm $\mathcal{A}$, and*

---

> *defender is a pair of algorithms $(\mathcal{T}, \Gamma)$, where $\mathcal{T}$ is a supervised learning algorithm, and $\Gamma$ is a test-time adaptation algorithm.*
>
> **Before game starts**
> - *A (labeled) training set $D$ is sampled i.i.d. from $(X^s, Y^s)$.*
>
> **Training time**
> - **(Defender, optional)** *Train a model $F$ trained on $D$ as $F = \mathcal{T}(D)$.*
>
> **Test time**
> - *A (labeled) natural test set $V$ is sampled i.i.d from $(X^t, Y^t)$, $V$ is sent to the attacker.*
> - **(Attacker)** *Produce an unlabeled dataset $U$ as follows:*
>
>     1. *On input $\Gamma$, $F$, $D$, and $V$, $\mathcal{A}$ perturbs each point $(x, y) \in V$ to $(x', y)$ (subject to the agreed attack type), giving $\widetilde{V} = \mathcal{A}(\Gamma, F, D, V)$.*
>     2. *Send $U = \widetilde{V}|_X$ (that is, the feature vectors of $\widetilde{V}$) to the defender.*
>
> - **(Defender)** *Produce an adapted model as $\widetilde{F} = \Gamma(F, D, U)$.*
>
> **Evaluation:** *Evaluate the test loss of $\widetilde{F}$ on $\widetilde{V}$, $L(\widetilde{F}, \widetilde{V})$.*
>
> **On the adversary:** *While the adversary knows the adaptation algorithm, $\Gamma$ may use some private randomness, such as random initialization. Since defender proceeds after the attacker to apply $\Gamma$, these private randomness is assumed not to be known by the adversary.*
>
> **On notations:** *If there is no need for a pretrained model, we use the notation $\Gamma(\bot, \cdot, \cdot)$.*

**Modeling differences.** **(1)** The first obvious distinction is that *source domain may differ from the target domain.* **(2)** This is called a *test-time* threat model since $\widetilde{F}$ is trained based on $U = \widetilde{V}|_X$, and the test error is only evaluated on $\widetilde{V}$ (the perturbed test set). This is in sharp contrast with the classic minimax threat model, where the model $F$ is trained on $D$, which is independent from $V$ and so $\widetilde{V}$. **(3)** This threat model enables us to study whether *a large test set (i.e. large $|U|$) can benefit a defender for adversarial robustness.* This is in sharp contrast with the classic minimax threat model, where the attacker is granted much power, and can pick points in a one-by-one, or "online", fashion (i.e., sample a point from the nature, perturb it, and send to the defender). **(4)** This is called a *maximin* threat model since the adversary *must move first* to submit the challenge set $U$, and the defender then moves based on it. In fact, this can be written as a maximin game $\mathbb{E}_V \left[ \max_U \min_{\widetilde{F}} \text{test\_err}(\widetilde{F}, \widetilde{V}) \right]$, different from the minimax game in the classic threat model.

**Same security goal.** However, in the case where these two domains coincide, the maximin threat model is still nontrivial. In fact, from the security perspective, both threat models are about the *same* end-to-end security goal: *Correct predictions on adversarially perturbed data.*

**Attacking a model vs. Attacking an algorithm.** In the classic threat model, a model $F$ is fixed after training, and at the test time the adversary attacks the model. However, in the maximin threat model, the adversary must attack an adaptation *algorithm* $\Gamma$.

**Restrictions of the threat model.** Astute readers may realize that we have intentionally left the definition of the test-time adaptation $\Gamma$ to have a lot of freedom. For example, $\Gamma$ can leverage the labeled training data $D$, which may be practically not available or computationally infeasible for test-time computations. We consider the following possibilities of restrictions:

***Homogeneity***. If the source and target domains equal, we call it the *homogeneous maximin model*. This setting is directly related to the classic threat model (Definition 1), where we have a single domain $(X, Y)$, but an adversary can perturb the test instances. From a security perspective, maximin threat model gives a *relaxed* threat model, but *the same end-to-end security guarantees*. (i.e., correct predictions against adversarially perturbed input)

***Data-obliviousness***. In this case, the adaptation algorithm $\Gamma$ cannot leverage the labeled source data $D$. In other words, the agent can only leverage the pretrained model $F$ and the unlabeled data set $U$.

### 3.2 ADVERSARIAL SEMI-SUPERVISED MINIMAX THREAT MODEL

An unsatisfying aspect of the theory thus far is that the classic minimax threat model is only defined in the homogeneous case, but the maximin threat model can be inhomogeneous, which makes the

direct comparison difficult. To bridge the gap, we provide a generalization of the classic minimax threat model that lies between the classic minimax threat model (Definition 1) and the test-time maximin threat model (Definition 2).

---

**Definition 3** (**Adversarial semi-supervised minimax threat model**). *Fix an adversarial perturbation type. We have a source domain $(X^s, Y^s)$, and a target domain $(X^t, Y^t)$. The attacker is a pair of algorithms $(\mathcal{A}_0, \mathcal{A}_1)$, and the defender is a pair of algorithms $(\mathcal{T}, \Gamma)$, where $\mathcal{T}$ is a supervised learning algorithm, and $\Gamma$ is a semi-supervised learning algorithm.*

**Before game starts**
- *A training set $D$ is sampled i.i.d. from $(X^s, Y^s)$.*
- *A semi-supervision set $V$ is sampled i.i.d from $(X^t, Y^t)$, $V$ is sent to the attacker.*

**Training time**
- **(Defender, optional)** *Train a model $F$ trained on $D$ as $F = \mathcal{T}(D)$.*
- **(Attacker)** *Produce an unlabeled dataset $U$ as follows:*

  1. *On input $\Gamma$, $F$, $D$, and $V$, $\mathcal{A}_0$ perturbs each point $(x, y) \in V$ to $(x', y)$ (subject to the agreed attack type), giving $\widetilde{V} = \mathcal{A}_0(\Gamma, F, D, V)$.*
  2. *Send $U = \widetilde{V}|_X$ (that is, the feature vectors of $\widetilde{V}$) to the defender.*

- **(Defender)** *Produce a final model as $\Gamma : \widetilde{F} = \Gamma(F, D, U)$.*

**Test time**
- *A (labeled) natural test set $V'$ is sampled i.i.d. from $(X^t, Y^t)$.*
- **(Attacker)** *On input $\widetilde{F}$, $D$, and $V'$, $\mathcal{A}_1$ perturbs each point $(x, y) \in V'$ to $(x', y)$ (subject to the agreed attack type), giving $\widetilde{V'} = \mathcal{A}_1(\widetilde{F}, D, V')$.*

**Evaluation:** *Evaluate the test loss of $\widetilde{F}$ on $\widetilde{V'}$, $L(\widetilde{F}, \widetilde{V'})$. The goal of the adversary $(\mathcal{A}_0, \mathcal{A}_1)$ is to (jointly) maximize $L(\widetilde{F}, \widetilde{V'})$.*

---

**Modeling differences.** **(1)** This threat model is *semi-supervised*, because the defender (learner) receives an unlabeled data set $U$ from the target domain. Note that the procedure to produce $\widetilde{F}$ at the training time of this threat model is exactly the same as the procedure to produce $\widetilde{F}$ in the test-time maximin threat model. The key difference is that, once $\widetilde{F}$ is trained, we evaluate it on $\widetilde{V'}$, which is the adversarially perturbed data on *independently sampled* $V'$. This is thus inductive learning, and a minimax game. **(2)** This threat model is *adversarial*, because the attacker can adversarially perturb the clean semi-supervision set $V$ to produce $U$.

**Classic minimax model as a special case.** The classic minimax threat model is a *special case* of this threat model, by putting source and target domains equal, and choosing a trivial $\Gamma$: $\Gamma(F, D, U) = F = \mathcal{T}(D)$. We list several other specializations of this threat model in Appendix B.3. Therefore, without loss of generality, for the rest of the paper, by *minimax model* we mean the *adversarial semi-supervised minimax threat model*.

$\Gamma$: **One algorithm, two interpretations.** We note that for an algorithm $\Gamma : (F, D, U) \mapsto \widetilde{F}$, one can have now two interpretations: In the maximin threat model, we interpret it as a adaptation algorithm, because we are only going to evaluate $\widetilde{F}$ on $\widetilde{V}$; in the minimax threat model, we interpret it as a semi-supervised learning algorithm, because we are going to evaluate $\widetilde{F}$ on unseen points $\widetilde{V'}$.

## 4 Separating Maximin and Minimax

We now move on to the focus of this work: *Is the maximin threat model "strictly weaker" than the minimax threat model?*

### 4.1 Valuation of the games

**Proposition 1** (**Homogeneous maximin vs. Classic minimax threat model**). *Let $k \geq 1$ be a natural number, and $\mathcal{F}$ be the hypothesis class. For a given $V$, the domain of $\widetilde{V}$ is a well-defined function of $V$ (e.g., $\ell_\infty$ ball around $V$). We have that:* $\mathbb{E}_{V \sim (X,Y)^k} \left[ \max_U \min_{\widetilde{F} \in \mathcal{F}} \{ L(\widetilde{F}, \widetilde{V}) \} \right] \leq \min_{\widetilde{F} \in \mathcal{F}} \mathbb{E}_{V \sim (X,Y)^k} \left[ \max_{\widetilde{V}} \{ L(\widetilde{F}, \widetilde{V}) \} \right]$

The proof ($A.1$) does not rely on the homogeneity condition, and holds verbatim to the more general semi-supervised threat model. We also note that, in fact, if the concept class has unbounded VC dimension, then good models always *exist* that can fit both $D$ and $V$ perfectly. So the valuation of the maximin game is actually always $0$:

**Proposition 2** (**Good models exist with large capacity**). *Consider binary classification tasks and that the hypothesis class $\mathcal{F}$ has infinite VC dimension. Then the valuation of the maximin game* $\mathbb{E}_{V \sim (X,Y)^k} \left[ \max_U \min_{\widetilde{F} \in \mathcal{F}} \{ L(\widetilde{F}, \widetilde{V}) \} \right]$ *is $0$. That is, perfect models always exist to fit $U$.*

This thus gives a first evidence that that transductive advesarial learning is strictly easier. We remark that transductive learning here is essential (differnet models are allowed for different $U$). We conclude this section by noting the following:

**Proposition 3** (**Good minimax solution is also a good maximin solution**). *Suppose $\mathcal{T}^*$ is a supervised learning algorithm which trains a model $F^* = \mathcal{T}^*(D)$, where its adversarial gain in the adversarial semi-supervised minimax model is bounded by $\kappa$ (i.e. $\mathbb{E}_{V'}[\max_{\widetilde{V}} L(F^*, \widetilde{V})] \le \kappa$.) Then in the maximin threat model, the adversarial gain of the strategy $(\mathcal{T}^*, \Gamma^*)$, where $\Gamma^*(F^*, D, U) = F^* = \mathcal{T}^*(D)$, is also upper bounded by $\kappa$.*

However, clearly, the valuation of the game does *not* answer a key question whether there is a "real" adaptation algorithm, which can only leverage unlabeled data $U$, that separates between the two theat models. In other words:

> *Is there a $\Gamma$ such that:*
> - *As a test-time adaptation algorithm in the maximin model, it provides robustness, but*
> - *As a learning algorithm in the minimax model, the model it produces has no robustness.*

The existence of such a $\Gamma$ would provide a separation between the minimax and the maximin threat models, and indicates that the maximin threat model admits more good solutions. Despite theoretical interest, this question is of practical importance since these "new" solutions may possess *desirable properties that good solutions in the minimax threat model may lack*. For the rest of this section, we consider one such desirable property that the defender solution is *attack agnostic* (Goodfellow (2018) (pp.30)): That is, The defender strategy is not to directly optimize the performance measure. (e.g., we know the attack is $\ell_\infty$-norm attacks, and we directly train for it).

## 4.2 PROVABLE SEPARATION OF THE MINIMAX AND MAXIMIN MODELS

In this section we provide a problem instance (i.e., data distributions and number of data points) and prove that that maximin threat model is *strictly easier* than the minimax threat model for the problem: In the minimax model no algorithm can achieve a nontrivial error, while in the maximin model there are algorithms achieving small errors. Since the maximin model is no harder than the minimax model for all problem instances and there is a problem instance where the former is strictly easier, we thus formally establish a separation between the two models. Furthermore, the problem instance we considered is on Gaussian data. The fact that maximin is already strictly easier than minimax in this simple problem provides positive support for potentially the same phenomenon on more complicated data.

**Data distributions and the learning task.** We consider the homogeneous case (the source and target are the same distribution) and the $\ell_\infty$ attack. We consider the Gaussian data model from Schmidt et al. (2018); Carmon et al. (2019b): a binary classification task where $\mathcal{X} = \mathbb{R}^d$ and $\mathcal{Y} = \{+1, -1\}$, $y$ uniform on $\mathcal{Y}$ and $x|y \sim \mathcal{N}(y\mu, \sigma^2 I)$ for a vector $\mu \in \mathbb{R}^d$ and coordinate noise variance $\sigma^2 > 0$. In words, this is a mixture of two Gaussians, one with label $+1$, and one with label $-1$. We will consider the following parameters. First, fix an integer $n_0 > 0$, and an $\epsilon \in (0, 1/2)$, then set the following parameter values:

$$\|\mu\|_2^2 = d, \quad \sigma^2 = \sqrt{d n_0}. \tag{1}$$

For both threat models, the datasets $D = \{(x_i, y_i)\}_{i=1}^n$ and $V = \{(x, y)\}$. In particular, $V$ only has one data point. In the maximin threat model, we let $x'$ denote the perturbed input obtained from $x$ by the $l_\infty$ attack with bounded norm $\epsilon > 0$, i.e., $x' = x + \nu$ with $\|\nu\|_\infty \le \epsilon$. Put $\widetilde{V} = \{(x', y)\}$ and $U = \{x'\}$. We prove the following:

**Theorem 1** (**Separation of Maximin and Minimax**). *Let $K$ be a (sufficiently large) constant (which is used for control classification error). Consider the Gaussian data model above with $n_0 \geq K$, $\sqrt{d/n_0} \geq \frac{32K \log d}{\epsilon^2}$, and $2Kn_0 \leq n \leq n_0 \cdot \frac{\epsilon^2 \sqrt{d/n_0}}{16 \log d}$ (the second holds for all sufficiently large $d$, which then determines all $n$ that falls into the range). Then the following are true.*

*(1) In the semi-supervised minimax threat model, the learned model $\widetilde{F} = \Gamma(\mathcal{T}(D), D, U)$ by any algorithms $\mathcal{T}$ and $\Gamma$ must have a large error: $\mathbb{E}\left\{ L(\widetilde{F}, \widetilde{V'}) \right\} \geq \frac{1}{2}(1 - d^{-1})$, where the expectation is over the randomness of $D, V$ and possible algorithm randomness.*

*(2) In the maximin threat model, there exist attack-agnostic algorithms $\mathcal{T}$ and $\Gamma$ such that for some absolute constant $c > 0$, the adapted model $\widetilde{F} = \Gamma(\mathcal{T}(D), D, U)$ has a small error: $\mathbb{E}\left\{ L(\widetilde{F}, \widetilde{V}) \right\} \leq e^{-cK}$.*

### 4.3 SEPARATION IN DEEP LEARNING: A CANDIDATE ALGORITHM OF DANN

We now consider deep learning. While we are not able to prove the existence of such a defender solution in this setting, we present a somewhat surprising connection between transductive adversarial robustness and unsupervised domain adaptation. In particular, we propose Domain Adversarial Neural Networks (DANN) as a *candidate algorithm* for the separation, and provide empirical evidence that it provides the desired separation. The experiment design is as follows:

**(1)** We use the DANN algorithm with random initialization (RI), that is, $\mathsf{DANN}(\mathsf{RI}, \cdot, \cdot)$, as a test-time adaptation. There are several motivations of choosing DANN: **(A)** DANN fits our framework as it is designed for unsupervised domain adaptation. **(B)** DANN is however *not* designed for adversarial robustness. Thus is it a very interesting question whether DANN can provide test-time adversarial robustness against attacks (e.g. norm-based attacks) that *it is not specifically designed for*. **(C)** DANN algorithm leverages source data $D$, which could benefit the maximin robustness.

**(2)** We generate adversarially perturbed $\widetilde{V}$, and check whether DANN can provide robustness.

**(3)** Note that $\mathsf{DANN}(\mathsf{RI}, \cdot, \cdot)$ can be viewed as a *semi-supervised learning algorithm in the adversarial semi-supervised minimax threat model*. Therefore, we check whether the adapted model $\widetilde{F} = \mathsf{DANN}(\mathsf{RI}, D, U)$ is robust in the minimax model.

**(4)** If (2) and (3) show that $\mathsf{DANN}(\mathsf{RI}, \cdot, \cdot)$ is *significantly more robust* in the test-time maximin threat model than in the minimax model, then the experiment goal is achieved.

**Attacks.** We use $\ell_\infty$ attacks in this section (note that DANN is not designed for $\ell_\infty$ attacks). We report more results about $\ell_2$ attacks in Appendix E.3. We consider two categories of attacks:

***Transfer attacks***. Transfer attacks are a common class of attacks, where we transfer attacks on one model to the target model (in our context, produced by a maximin defender strategy). In this paper we will mainly apply PGD attacks on adversarially trained models to generate transfer attacks.

***Adaptive attacks***. We also consider *adaptive attacks*[4], where an adversary can leverage the knowledge of the adaptation algorithm. To this end, we notice that test-time adaptation is typically an optimization objective $\mathcal{L}_{\mathrm{tta}}(\widetilde{F}, F, D, U)$, which gives rise to the following bilevel optimization objective for the attacker:

$$
\begin{aligned}
\underset{\widetilde{V} \in N(V)}{\text{maximize}} \quad & \mathcal{L}(\widetilde{F}^*, F, D, \widetilde{V}) \\
\text{subject to: } & \widetilde{F}^* \in \underset{\widetilde{F}}{\arg\min} \, \mathcal{L}_{\mathrm{tta}}(\widetilde{F}, F, D, U = \widetilde{V}|_X)
\end{aligned}
\tag{2}
$$

To solve this bilevel optimization, we generalize the work of Lorraine & Duvenaud (2018) to an algorithmic framework, called Fixed Point Alternating Method (Algorithm 1 and 2). We consider two instantiations in this section: **(1)** $\mathsf{FPAM}^L_{\mathcal{L}_{\mathsf{DANN}}}$, which is a standard instantiation, where the outer objective is $L(\widetilde{F}, \widetilde{V})$, and **(2)** $\mathsf{J\text{-}FPAM}^{\mathcal{L}_{\mathsf{DANN}}}_{\mathcal{L}_{\mathsf{DANN}}}$, where the outer objective is the exact same DANN objective $\mathcal{L}_{\mathsf{DANN}}$. This is also a natural instantiation where the adversary sets out to fail the DANN

---

[4]Please refer to Section D for derivations.

optimization. Note that in this case one can naturally apply the more traditional alternating optimization J-FPAM, because the objective becomes a more traditional minimax objective.

**Datasets.** For the homogeneous case, we consider MNIST and CIFAR10 (i.e., for both source and target domains). The homogeneous case represents the security problem considered in the classic threat model. For the inhomogeneous case, we consider MNIST→MNIST-M (Ganin et al., 2017), and CIFAR10→CIFAR10c-fog (Hendrycks & Dietterich, 2019). MNIST-M is a standard dataset in unsupervised domain adaptation, and CIFAR10c is a recent benchmark for evaluating neural network robustness against common corruptions and perturbations. For CIFAR10c-fog, all 5 levels are combined and we perform an 80-20 train-test split, which gives a training set of size 40000 and test set of size 10000. (for space reasons, we combine all corruption levels for experiments in this section, results where we separately study different levels are reported in Section E.2).

**Models.** For MNIST, we use the original DANN architecture from Ganin et al. (2017) with slight modifications (e.g. adding batch normalization and dropout layers). For CIFAR10, we use pre-activation ResNets from He et al. (2016) for the prediction branch of DANN, and for the domain prediction branch we use the architecture from Sun et al. (2020) (convolution layer). This architecture is slightly different from the the typical DANN architecture used for MNIST, which assumes vectorized feature and fully connected domain prediction branch.

| Homogeneous ($S = T$) | | | Inhomogeneous ($S \neq T$) | | |
|---|---|---|---|---|---|
| **Data** | **Defender** | **Accuracy** | **Data** | **Defender** | **Accuracy** |
| MNIST | DANN | 98.79% | MNIST→MNIST-M | DANN | 62.73% |
| MNIST | AdvS | 98.14% | MNIST→MNIST-M | AdvS | 35.95% |
| CIFAR10 | DANN | 49.47% | CIFAR10→CIFAR10c-fog | DANN | 52.28% |
| CIFAR10 | AdvS | 46.39% | CIFAR10→CIFAR10c-fog | AdvS | 17.22% |

Table 1: **Adversarial robustness under transfer attacks with $\ell_\infty$ attacks**. For MNIST we use perturbation budget $0.3$. For MNIST-M we use perturbation budget $8/255$ (we found robustness degrades fast on MNIST-M, even for adversarially trained models, so we choose this number to illustrate relative performance). For all CIFAR experiments we use perturbation budget $8/255$. For transfer attacks, DANN remained robust (albeit in a weaker threat model). Note that the numbers on adversarial training are only provided to qualitatively indicate the robutness level of DANN. The main goal is to provide evidence for separation.

**Experiment results.** **(A)** As a sanity check first, we evaluate the accuracy of the adapted DANN models in the minimax threat model. Not surprisingly, the accuracy becomes very low (close to $0\%$),

| $k$ | MNIST | MNIST→MNIST-M | CIFAR10 | CIFAR10→CIFAR10c-fog |
|---|---|---|---|---|
| 1 | 98.57% | 56.37% | 63.65% | 67.99% |
| 2 | 98.85% | 55.62% | 83.62% | 56.06% |
| 3 | 99.20% | 55.54% | 83.52% | 50.12% |
| 4 | 99.42% | 52.87% | 88.01% | 49.52% |
| 5 | 98.23% | 52.31% | 88.18% | 55.09% |
| 6 | 98.71% | 53.49% | 88.52% | 57.25% |
| 7 | 99.35% | 52.57% | 88.24% | 55.85% |
| 8 | 98.69% | 52.16% | 87.91% | 54.89% |

Table 2: **Adversarial robustness under** $\mathsf{FPAM}^L_{\mathcal{L}_{\mathsf{DANN}}}$ **adaptive attacks, with** $F = \mathsf{AdvT}$ (i.e., the attacker initializes the initial model as the adversarially trained target model). DANN exhibits significant robustness even for large $k$.

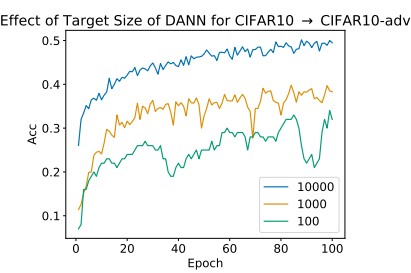

Effect of Target Size of DANN for CIFAR10 → CIFAR10-adv

Figure 1: **Effect of test size on CIFAR10.** We plot accuracy as we train more epochs with DANN during the adaptation, for *different* target size. Adversarial accuracy of smaller target size gets lower accuracy.

| Homogeneous ($S = T$) | | | Inhomogeneous ($S \neq T$) | | |
|---|---|---|---|---|---|
| **Data** | **Attacker** | **Accuracy** | **Data** | **Attacker** | **Accuracy** |
| CIFAR10 | J-FPAM$^{\mathcal{L}}_{\mathcal{L}_{\mathsf{DANN}}}$ | 42.22% | CIFAR10→CIFAR10c-fog | J-FPAM$^{\mathcal{L}}_{\mathcal{L}_{\mathsf{DANN}}}$ | 31.70% |

Table 3: **Adversarial robustness under** J-FPAM$^L_{\mathcal{L}_{\mathsf{DANN}}}$ **adaptive attacks for CIFAR10 and CIFAR10c-fog tasks.** For FPAM$^L_{\mathcal{L}_{\mathsf{DANN}}}$ we use $k = 20$ (early stopping according to Table 2). For J-FPAM$^{\mathcal{L}}_{\mathcal{L}_{\mathsf{DANN}}}$, we use $k = 100$. Joint FPAM is by far our most effective for DANN, though DANN still achieves non-trivial test time robustness, and demonstrate a separation between maximin and minimax threat models.

which shows that DANN provides no robustness in the minimax threat model. **(B)** Then in the maximin threat model, Table 1 summarizes the results under transfer attacks. In the homogeneous case, the adversarial accuracy DANN provided in the maximin model is comparable to the adversarial accuracy an adversarially trained model provided in the minimax model. And the adversarial accuracy becomes significantly higher in the inhomogeneous case (compared to using an adversarially trained source model). **(C)** In the maximin threat model, Table 2 summarizes the results under $\text{FPAM}^L_{\mathcal{L}_{\text{DANN}}}$ attacks. Similar to the transfer attack case, DANN provides noticeable robustness. Note that since the defender moves after the attacker, he or she always applies adaptation "one more time". **(D)** In the maximin threat model, Table 3 summarizes the results under $\text{J-FPAM}^L_{\mathcal{L}_{\text{DANN}}}$ attacks. This is by far our most effective attacks against DANN, which decreases the robustness to $\sim 40\%$ in the homogeneous case on CIFAR10, and to $\sim 30\%$ in the inhomogenous case on CIFAR10→CIFAR10c-fog. Nevertheless, DANN still provides nontrivial robustness, and thus provides positive evidence to our hypothesis that DANN separates maximin and minimax threat models. **(E)** Finally, Figure 1 gives the robustness results under different target size. As we can see, the robustness of DANN degrades as the target size decreases, confirming our intuition that large target size benefits test-time robustness.

## 5 ROBUSTNESS OF OBLIVIOUS ADAPTATION AND FUTURE DIRECTIONS

We briefly explore the adversarial robustness of the recent data-oblivious test-time adaptation algorithms. Specifically, we focus on the TTT algorithm by Sun et al. (2020). Recall that a test-time adaptation algorithm is data-oblivious, if it does not use the labeled training data $D$ at the test time. TTT algorithm uses a specially trained (with self training) pretrained model, which we denote as PrTTT. Similar to our DANN experiment, we conduct experiments on CIFAR10→CIFAR10c-fog: **(1)** We found that *transfer attacks against* PrTTT can already break TTT algorithm (close to $0\%$ accuracy). While this is not surprising as authors of Sun et al. (2020) have cautioned that TTT is not designed for adversarial robustness, this is in sharp contrast to our results with DANN. **(2)** We then use an adversarially trained model as the pretrained model for TTT. We found that this indeed increases the maximin-robustness of TTT. However, the robustness is roughly the same (about $1\%$ difference) as directly using the pretrained model. This indicates that the robustness mainly comes from the adversarially trained model, and TTT algorithm provides no robustness. This is again in sharp contrast to DANN. We list several implications as future directions:

• *Is transductive adversarial deep learning easier?* Both the current work and the work of Goldwasser et al. (2020) have given indication that adversarial robustness may be easier in the transductive learning setting. To this end, we ask: Can we devise more effective algorithms against DANN, or we can prove its robustness in the maximin threat model? To this end, either confirming (theoretically, for example) or refuting the maximin robustness of DANN is intriguing. To make this concrete, we propose the following (informal, qualitative) conjecture:

**Conjecture 1** (**Transductive robustness of DANN (informal, qualitative)**). *If* DANN *can successfully adapt from source domain* $(X^s, Y^s)$) *to* $(X^t, Y^t)$ *(no adversary), then* DANN *can provide test-time maximin robustness for bounded norm attacks on* $(X^t, Y^t)$.

Specifically, for a successful refutation, one must provide two (natural) domains where DANN can successfully adapt from the source to the target , but DANN fails to provide nontrivial transductive robustness for bounded norm attacks on the target domain.

• *The surprising power of distribution matching for robustness.* It is somewhat surprising that a distribution matching algorithm, such as DANN, can provide adversarial robustness (albeit in a weaker threat model), even when the target domain is already a corrupted domain such as CIFAR10c-fog. This means that for practical applications where we run an ML model in a batch mode on some collected unlabeled data, one can consider first apply UDA to produce an adapted model and then classify the batch, and this may provide adversarial robustness.

• *Does robust test-time adaptation always need to access source data?* We note that while DANN provides maximin robustness, it needs to access the labeled training data $D$ at test time, and while TTT does not need $D$, it provides no maximin robustness. Can we achieve the best of both of worlds and get adversarially robust *oblivious* test-time adaptation algorithms?

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

# A    PROOFS

## A.1    PROOF OF PROPOSITION 1

Let $\mathcal{F}$ be the family of models $\widetilde{F}$ we can choose from. From the maximin inequality, we have that

$$\max_{U} \min_{\widetilde{F} \in \mathcal{F}} \{L(\widetilde{F}, \widetilde{V})\} \leq \min_{\widetilde{F} \in \mathcal{F}} \max_{\widetilde{V}} \{L(\widetilde{F}, \widetilde{V})\}$$

Note that for the minimax, the max over $\widetilde{V}$ is also constrained to perturb features (as we want to find adversarial examples). If we take expectation over $V$, we have then

$$\mathbb{E}_{V} \left[ \max_{U} \min_{\widetilde{F} \in \mathcal{F}} \{L(\widetilde{F}, \widetilde{V})\} \right] \leq \mathbb{E}_{V} \left[ \min_{\widetilde{F} \in \mathcal{F}} \max_{\widetilde{V}} \{L(\widetilde{F}, \widetilde{V})\} \right]$$

Note that

$$\mathbb{E}_{V} \left[ \min_{\widetilde{F} \in \mathcal{F}} \max_{\widetilde{V}} \{L(\widetilde{F}, \widetilde{V})\} \right] \leq \min_{\widetilde{F} \in \mathcal{F}} \mathbb{E}_{V} \left[ \max_{\widetilde{V}} \{L(\widetilde{F}, \widetilde{V})\} \right],$$

which completes the proof.    □

## A.2    PROOF OF PROPOSITION 3

In the maximin threat model, let us take the strategy $(\mathcal{T}^*, \Gamma^*)$ as defined. Therefore the maximin adversarial gain is

$$\mathbb{E}_{V} \left[ \max_{\widetilde{V}} L(\Gamma^*(\mathcal{T}^*(D), D, \widetilde{V}|_X), \widetilde{V}) \right] = \mathbb{E}_{V} \left[ \max_{\widetilde{V}} L(\mathcal{T}^*(D), \widetilde{V}) \right] = \mathbb{E}_{V} \left[ \max_{\widetilde{V}} L(F^*, \widetilde{V}) \right]$$

which, by assumption, is bounded by $\kappa$.    □

# B    MORE ON THREAT MODELS

## B.1    ON THE MODELING

**Why is the adversary constrained to attack each point in Definition 2 and Definition 3?** This is set up as the rule of the threat model, because if not, an adversary can pick any point that fails the model and repeats it arbitrary number of times, which trivializes the model. In fact, such one-by-one attack is implicit in the original minimax objective, and is also the common practice of evaluating adversarial robustness (i.e., typical evaluation is to take a test set, attack the points one by one, and then evaluate adversarial accuracy). Our threat model formulation makes this hidden assumption explicit. We refer readers to Goodfellow (2019) for more discussion if one wants to break the i.i.d. sampling assumption.

**Why in the minimax modeling, we do not assume $\mathcal{A}$ in Definition 1, or $\mathcal{A}_1$ in Definition 3 to know the training algorithms?** For example, $\mathcal{T}$ is not in the input of $\mathcal{A}$ in Definition 1. This is the case because in the inner most expression $L(F, \widetilde{V})$ only depends on $F$, and $F$ is fixed after the training. Therefore, bringing in the training algorithm gives no additional information (note that this is also the typical practice of evaluating adversarial robustness in the classic setting, where we just attack the model). By contrast, this is not the case for the maximin threat model because attacker moves first and the model is not fixed yet, so in that case, a strong white-box attacker may need to take into consideration the adaptation algorithm, not just the pretrained model.

**Why do we need to have two algorithms $\mathcal{A}_0$, $\mathcal{A}_1$ in Definition 3?** $\mathcal{A}_0$ attacks the semi-supervised learning algorithm, and thus has $\Gamma$ as input. On the other hand, $\mathcal{A}_1$ attacks a model, so neither $\Gamma$ nor $\mathcal{T}$ is in the input. Therefore, these two algorithms are different and are separated in the definition. Note that the joint goal of $\mathcal{A}_0$, $\mathcal{A}_1$ is to fail the final model on $\widetilde{V'}$, so $\mathcal{A}_0$ can be totally different from $\mathcal{A}_1$ in order to fail the training. In this respect, $\mathcal{A}_0$ is like data poisoning the unlabeled data set in the semi-supervised learning. Unlike data poisoning though, we have the constraint that it must attack each point in $V$.

**There are two algorithms $\mathcal{A}_0$, $\mathcal{A}_1$ in Definition 3, wouldn't $\mathcal{A}_0$ help the defender because it leaks information about the target distribution.** We have shown that the defender can choose to ignore the unlabeled data set, and this degenerates to the classic threat model in the homogeneous case. Otherwise, the semi-supervised learning setting may indeed help the defender, because the information about the target domain (or more unlabeled data if we are in the homogeneous case) is leaked to the defender. This is, however, exactly why in these cases that this threat model lies between the classic threat model and the maximin threat model. In the maximin threat model, the attacker leaks $U$ to the defender, and the defender is only required to predict $U$. In the semi-supervised threat model, things becomes more challenging, because finally we want to evaluate the model on independent samples.

**In the experiments, we essentially instantiated the semi-supervised threat model, for example, as $(\mathcal{A}_0 = \mathsf{FPA}, \mathcal{A}_1 = \mathsf{PGD})$. Is that too weak?** A strong adversary of course may want to use a strong attack $\mathcal{A}_0$ to fail the training, and thus fail the final model, and choosing FPA may indeed be a weak choice. However, we note that our goal in Step (3) of the experiment design is to show that we can *break* the adapted model, and $(\mathsf{FPA}, \mathsf{PGD})$ already full-fills this purpose. As another note, in the maximin threat model, currently FPA is the most natural adaptive attack we can think of. Exploring stronger attacks is an intriguing future direction.

## B.2 MORE EXTENSIONS AND RESTRICTIONS FOR MAXIMIN THREAT MODEL

***Singleton.*** In the standard model, we allow $U$ to be a batch which consists of many point. In the singleton restriction, $U$ is restricted to have only one point (for example, the test-time training algorithm by Sun et al. (2020) works in this setting).

**Definition 4** (**Online version**). *In the online maximin model, we assume that we will sequentially receive batches of test samples, one after another, and we allow $\Gamma$ to leverage the previous samples received. We assume, for simplicity of this work however, that the batches are sampled independently from each other.*

For the online maximin model, data obliviousness means that the agent cannot directly leverage the previous unlabeled samples he or she received. However, the agent can save the previous model parameters and use that.

## B.3 SPECIALIZATION OF THE ADVERSARIAL SEMI-SUPERVISED MINIMAX THREAT MODEL

**Adversarial training considered in Miyato et al. (2019); Shu et al. (2018).** The setting becomes a special case of our general minimax threat model by:

- Instantiating $\mathcal{A}_0$ as a trivial adversary that does not do any perturbation.That is the defender receives a set of clean set $U$ of unlabeled features.

- The defender adaptation strategy does adversarial training on the unlabeled data, given by

$$d(p(y|x), p(y|x + r^*, \theta))$$
$$\text{where } r^* = \underset{r:\|r\| \leq \varepsilon}{\arg\max} \, d(p(y|x), p(y|x + r, \theta))$$

**Unsupervised Adversarial Training** Uesato et al. (2019) This is similar to the above except that the source and target domains are equal.

## C  RELATED WORK

**Adversarial robustness.** Adversarial robustness of deep learning has received significant attention in recent year. By far a standard approach is to do adversarial training directly for the attack type Madry et al. (2018); Sinha et al. (2018). Several recent works have studied adversarial training in a semi-supervised setting Carmon et al. (2019a); Uesato et al. (2019). As we have discussed, these work can be viewed as instantiations in the adversarial semi-supervised minimax threat model. Our work can be viewed as one step further to study adversarial robustness in the transductive setting for deep learning.

**Test-time adaptation and transductive learning.** Transductive learning is a long line of research (for example, see the classic work of Transductive Support Vector Machine Joachims (1999)), where one attempts to leverage unlabeled data $U$ to predict on specific instances of $U$ Vapnik (1998). On the other hand, transductive learning has not received much attention in the deep learning setting. The recent proposals of test-time adaptation Sun et al. (2020); Wang et al. (2020); Nado et al. (2020) seems to be an reincarnation of this idea, with the additional twist of leveraging a pretrained model. Our work considers test-time adaptation in an adversarial deep learning setting, which to the best of our knowledge, is new.

**Unsupervised Domain Adaptation.** A classic approach for analyzing domain adaption is based on $\mathcal{H}$-divergence Kifer et al. (2004); Blitzer et al. (2008); Ben-David et al. (2010). That theoretical framework is the basis for a line of methods that uses adversarial training with neural networks to learn representations that are indistinguishable between source and target domain, in particular domain adversarial neural network (DANN) Ajakan et al. (2014); Ganin et al. (2017) and related techniques Pei et al. (2018); Zhao et al. (2018). Some other approach used different divergence notions, such as MMD Long et al. (2014; 2015), Wasserstein distance Courty et al. (2017); Shen et al. (2018), and Rényi divergence Mansour et al. (2009).

## D  ADAPTIVE ATTACKS AND BILEVEL OPTIMIZATION

In this section we present details of our considerations of adaptive attacks in the test-time adaptation setting. To start with, in the maximin threat model, for a fixed $D$ (labeled training set) and $V$ (labeled test set), the game is $\max_U \min_{\widetilde{F}} L(\widetilde{F}, \widetilde{V})$ (the adversary can only modify features, namely $U$, without modifying the labels). We focus on norm-bounded attacks, even though the test-time defense strategy can be agnostic to the attack type (i.e., the adaptation algorithm does not explicit leverage the attack type information). More specifically, the constraint of the adversary is that he can only perturb the feature of each point in $V$ using norm-based perturbation: Namely for each $(x, y)$ the adversary can generate $(x', y)$ where $\|x - x'\| \leq \varepsilon$. we use the notation $N(V)$ to denote the neighborhood of $V$ which includes all such legitimate perturbed datasets.

In our settings, the adaptation algorithm $\Gamma$ is typically an optimization objective (which revises the model, we will demonstrate an instantiation using DANN below). We assume that this objective is $\mathcal{L}_{\text{tta}}(\widetilde{F}, F, D, U)$, where $\widetilde{F}$ is the we want to solve by optimization, $F$ is a (fixed) pretrained model (which can be $\perp$ if a pretrained model is not needed, see the DANN instantiation below), $D$ is the labeled data set from source domain, and $U = \widetilde{V}|_X$ is the test features (of $\widetilde{V}$) from the target domain. Under these conditions, the maximin game can then be written as a *bilevel optimization* as follows:

$$\begin{aligned} \underset{\widetilde{V} \in N(V)}{\text{maximize}} \quad & L(\widetilde{F}^*, \widetilde{V}) \\ \text{subject to: } & \widetilde{F}^* \in \arg\min_{\widetilde{F}} \mathcal{L}_{\text{tta}}(\widetilde{F}, F, D, U = \widetilde{V}|_X) \end{aligned} \tag{3}$$

To incorporate more objectives, such as that the adversary can attack the inner minimization as a special case, we generalize the loss function of the outer optimization to the form of $\mathcal{L}(\widetilde{F}, F, D, V)$. Note that compared to $\mathcal{L}_{\text{tta}}$ we allow labeled data $V$ for the attacker). This gives rise to the following objective (we have specifically generalized the outer maximization objective):

$$\begin{aligned} \underset{\widetilde{V} \in N(V)}{\text{maximize}} \quad & \mathcal{L}(\widetilde{F}^*, F, D, \widetilde{V}) \\ \text{subject to: } & \widetilde{F}^* \in \arg\min_{\widetilde{F}} \mathcal{L}_{\text{tta}}(\widetilde{F}, F, D, U = \widetilde{V}|_X) \end{aligned} \tag{4}$$

In words, the outer optimization is the adversarial game where the adversary tries to find $\widetilde{V}$ to fool the defense, while the inner minimization solves the test-time adaptation objective $\mathcal{L}_{\text{tta}}$, leveraging the information of $U = \widetilde{V}|_X$, in order to derive a revised model $\widetilde{F}^*$ for predictions.

**Example: An objective of maximin game with DANN as an defender**. If the inner test-time defense DANN, then we know that $\widetilde{F}$ can be written as $f \circ \phi$, and we can write the following bilevel optimization objective, where the inner minimization is the DANN objective Ganin et al. (2017):

$$\begin{aligned} \underset{\widetilde{V} \in N(V)}{\text{maximize}} \quad & L(\widetilde{F}, \widetilde{V}) \\ \text{subject to: } & \widetilde{F} \in \arg\min_{\phi, f} \left\{ L(f \circ \phi, D) + d\left( \phi(D), \phi(\widetilde{V}|_X) \right) \right\} \end{aligned}$$

where $d$ is a distance function, which is realized as a domain discriminator (network). We refer readers to Ganin et al. (2017) for more details. We remark that this objective with DANN gives evidence that: **(1)** Defenses using test-time adaptation are significantly different from test-time defenses discussed in Athalye et al. (2018). **(2)** The game is harder for the adversary because he or she needs to solve a harder optimization problem. To this end, we note that the test-time defenses discussed in Athalye et al. (2018) do not amount to bilevel optimizations. This is because that those defenses are merely about sanitizing input $x$ given a *fixed* pretrained model, which is known to the adversary at the attack time and can thus be differentiated. On the other hand, the use of DANN as a test-time adaptation algorithm makes bilevel optimization essential to the adversarial game.

---

**Algorithm 1** FIXED POINT ALTERNATING METHOD $\text{FPAM}_{\mathcal{L}_{\text{tta}}}^{\mathcal{L}}[k, F]$

---

**Require:** A training dataset $D$, a natural test set $V$, an (optional) pretrained model $F$ for test-time adaptation, and an integer parameter $k \geq 0$ (the number of rounds).
1: If the pretrained model $F$ equals $\perp$, set $U_0 = V|_X$. Otherwise, attack the pretrained model $F$ on $V$, by fixing $F$ and solves the objective $\max_{\widetilde{V} \in N(V)} L(F, \widetilde{V})$ (i.e., the standard test loss of $F$), to get adversarially perturbed examples $V_0$. Set $U_0 = V_0|_X$.
2: **for** $i = 1, 2, \ldots, k$ **do**
3:    Solve the inner minimization objective $F_i = \arg\min_{\widetilde{F}} \mathcal{L}_{\text{tta}}(\widetilde{F}, F, D, U_{i-1})$.
4:    Solve the outer maximization objective $V_i = \text{argmax}_{\widetilde{V} \in N(V_{i-1})} \mathcal{L}(F_i, F, D, \widetilde{V})$. Set $U_i = V_i|_X$.
5: **end for**
6: **return** $U_k$.

---

**Solving the bilevel optimization (4).** To solve the bilevel optimization, we propose two algorithm frameworks. Fixed Point Alternating Method (FPAM, Algorithm 1), and Joint Fixed Point Alternating Method (J-FPAM, Algorithm 2). These two algorithms generalize the work of Lorraine & Duvenaud (2018) (specifically, their Algorithm 2, "optimization of hypernetwork, then hyperparameters"), to solve (4). Specifically, the joint optimization can be effective in the case where the outer and inner objectives are similar to each other, in which case the objective degnerates to a more traditional minimax objective for optimization.

---

**Algorithm 2** JOINT FIXED POINT ALTERNATING METHOD J-FPAM$_{\mathcal{L}_{\text{tta}}}^{\mathcal{L}}[k, F]$

---

**Require:** A training dataset $D$, a natural test set $V$, an (optional) pretrained model $F$ for test-time adaptation, and an integer parameter $k \geq 0$ (the number of rounds).

1: If the pretrained model $F$ equals $\perp$, set $U_0 = V|_X$. Otherwise, attack the pretrained model $F$ on $V$, by fixing $F$ and solves the objective $\max_{\widetilde{V} \in N(V)} L(F, \widetilde{V})$ (i.e., the standard test loss of $F$), to get adversarially perturbed examples $V_0$. Set $U_0 = V_0|_X$.
2: **for** $i = 1, 2, \ldots, k$ **do**
3:     **for** minibatch $V_B \subset V$ **do**
4:         Perform PGD on the outer maximization objective: $\widetilde{V} = \text{argmax}_{\widetilde{V} \in N(V_{i-1})} \mathcal{L}(F_i, F, D, \widetilde{V})$.
5:         Set $\widetilde{U} = \widetilde{V}|_X$.
6:         Perform an SGD step on the inner minimization objective: $F = F - \alpha \nabla \mathcal{L}_{\text{tta}}(\widetilde{F}, F, D, \widetilde{U})$.
7:     **end for**
8: **end for**
9: **return** $U_k$.

---

**Instantiations of** FPAM$_{\mathcal{L}_{\text{tta}}}^{\mathcal{L}}$ **with DANN.** We now instantiate this framework by considering $\mathcal{L}_{\text{tta}}$ as the DANN objective (with random initialization). In this case, there is no pretrained model $F$, so the inner test-time adaptation objective simplifies to $\mathcal{L}_{\text{DANN}}(\widetilde{F}, D, U)$. We consider two instantiations of the outer maximization:

FPAM$_{\mathcal{L}_{\text{DANN}}}^{L}$**: *Outer objective is* $L(\widetilde{F}, \widetilde{V})$.** This is the most standard instantiation, where for the outer maximization, the adversary directly search for $\widetilde{F}$ to maximize the test loss of $\widetilde{F}$ on $\widetilde{V}$.

J-FPAM$_{\mathcal{L}_{\text{DANN}}}^{\mathcal{L}_{\text{DANN}}}$**: *Outer objective is* $\mathcal{L}_{\text{DANN}}$, *the same* DANN *objective*.** This is a natural and standard instantiation where the adversary uses the same DANN objective for both inner and outer objectives. In this case, J-FPAM can be naturally applied as an alternating optimization based solver.

# E   MORE EXPERIMENTS

## E.1   BASELINES: ACCURACY OF ADVERSARIALLY TRAINED MODELS.

| Data | Attack distance | Training PGD steps ($\mathcal{T}$) | Test PGD steps ($\mathcal{A}$) | Nat. test acc | Adv. test acc |
|---|---|---|---|---|---|
| CIFAR10 | 8/255 | 7 | 20 | 78.31% | 46.39% |
| CIFAR10-fog | 8/255 | 7 | 20 | 71.39% | 29.1% |
| MNIST | 0.3 | 40 | 100 | 98.81% | 98.14% |
| MNIST-M | 8/255 | 40 | 100 | 96.50% | 80.88% |

Table 4: Accuracy of adversarially trained models in the minimax threat model. For the Nat. test acc column we do not use the adversary $\mathcal{A}$. Note that for the MNIST-M row it is the case that we directly do adversarial training on the clean labeled target data. This cannot be achieved in our threat model because defender only sees perturbed unlabeled target data. Nevertheless, this row gives the best adversarial accuracy one can hope for.

See Table 4 for the results.

## E.2   TRANSFER ATTACKS ON DIFFERENT CORRUPTION LEVELS

| Corruption level | AdvS on PGD(AdvS, CIFAR10c-fog) | AdvT on PGD(AdvT, CIFAR10c-fog) | DANN on PGD(AdvT, CIFAR10c-fog) | DANN on CIFAR10c-fog |
|---|---|---|---|---|
| 1 | 37.99% | 47.00% | 70.73% | 76.32% |
| 2 | 23.07% | 39.84% | 63.05% | 77.00% |
| 2 | 13.72% | 30.07% | 59.01% | 75.56% |
| 4 | 8.45% | 21.41% | 52.93% | 74.41% |
| 5 | 3.37% | 10.58% | 40.58% | 62.79% |

Table 5: Transfer attacks ($\ell_\infty$ attack) with different corruption levels.

See Table 5 for the results.

## E.3 TRANSFER ATTACKS WITH $\ell_2$ ATTACKS

| Homogeneous ($S = T$) | | | Inhomogeneous ($S \neq T$) | | |
|---|---|---|---|---|---|
| **Data** | **Defender** | **Accuracy** | **Data** | **Defender** | **Accuracy** |
| MNIST | DANN | 99.19% | MNIST→MNIST-M | DANN | 68.37% |
| MNIST | AdvS | 99.07% | MNIST→MNIST-M | AdvS | 44.30% |
| CIFAR10 | DANN | 69.57% | CIFAR10→CIFAR10c-fog | DANN | 69.98% |
| CIFAR10 | AdvS | 70.97% | CIFAR10→CIFAR10c-fog | AdvS | 37.32% |

Table 6: Transfer attacks ($\ell_2$ attacks with $\epsilon = 80/255$) for all tasks.

See Table 6 for the results.

**Plots of fixed point attacks** See Figure 2 for the results.

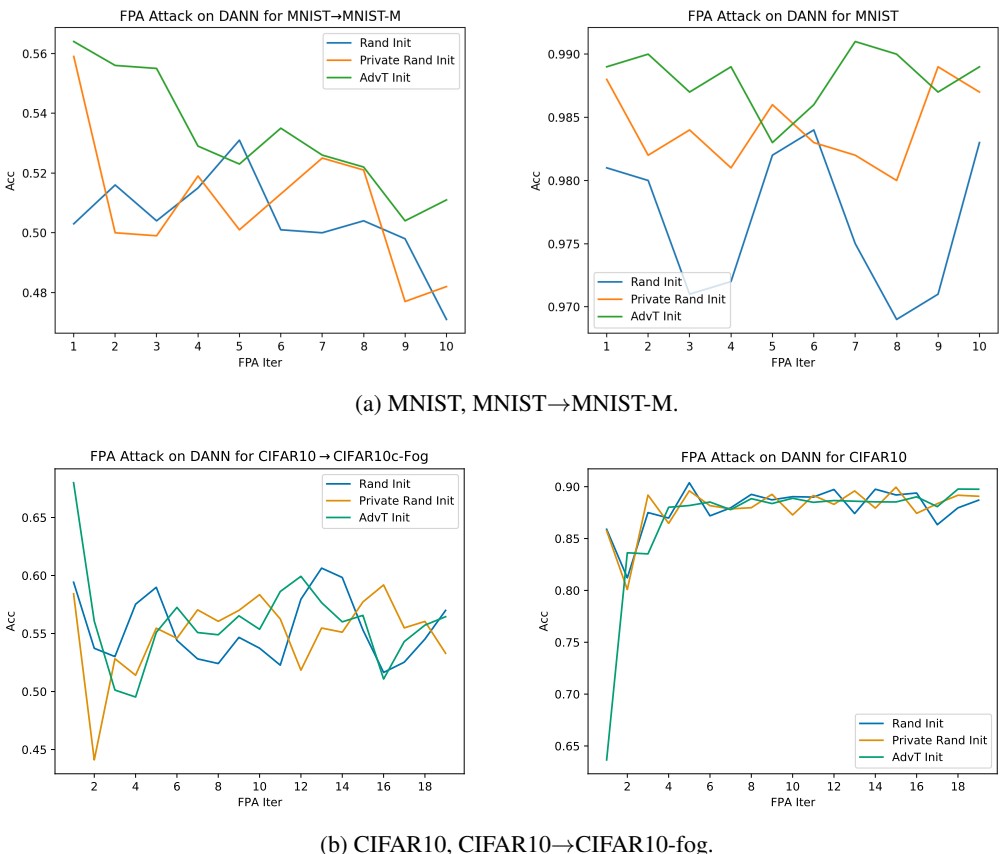

(a) MNIST, MNIST→MNIST-M.

(b) CIFAR10, CIFAR10→CIFAR10-fog.

Figure 2: Plots for fixed point attacks.

## E.4 DETAILS ON EXPERIMENT

**Models.** For MNIST, we use the original DANN architecture from Ganin et al. (2017) with slight modifications (e.g. adding batch normalization and dropout layers). For CIFAR10, we use pre-activation ResNets from He et al. (2016) for the prediction branch of DANN, and for the domain prediction branch we use the architecture from Sun et al. (2020) (convolution layer). This architecture is slightly different from the the typical DANN architecture used for MNIST, which assumes vectorized feature and fully connected domain prediction branch.

**Training Details.** For our CIFAR10 and CIFAR10c experiment, we obtain both the baseline standard pretrained model and the adversarial pretrained model via the following optimization schemes: SGD with 150 epochs, multiple step learning rate $[0.1, 0.01, 0.001]$ with milestone at epoch $[75, 125]$, momentum 0.9, and weight decay 0.0005, and batch size 128. The baseline adversarial pretrained model is trained with 7-step PGD. In evaluating the test time robustness for transfer attacks, we generate the adversarial samples with 20-step PGD. In evaluating the test time robustness for adaptive attacks, we evaluate with 7-step PGD.

For our MNIST and MNIST-M experiment, we obtain both the baseline standard pretrained model and the adversarial pretrained model via the following optimization schemes: ADAM with 100 epochs, learning rate $3 \times 10^{-4}$ and batch size 128. The baseline adversarial pretrained model is trained with 40-step PGD. In evaluating the test time robustness for transfer attacks, we generate the adversarial samples with 100-step PGD. In evaluating the test time robustness for adaptive attacks, we evaluate with 40-step PGD.

The optimization scheme for DANN during test time adaptation (both transfer and adaptive attack experiments) follows as: ADAM with learning rate 0.004, batch size 128.

## F   PROOF OF THEOREM 1

We prove this theorem by a series of lemmas.

**Lemma 1 (Part (1)).** *In the semi-supervised minimax threat model, the learned model $\widetilde{F} = \Gamma(\mathcal{T}(D), D, U)$, by any algorithms $\mathcal{T}$ and $\Gamma$, must have a large error:*

$$\mathbb{E}\left\{L(\widetilde{F}, \widetilde{V'})\right\} \geq \frac{1}{2}(1 - d^{-1}), \tag{5}$$

*where the expectation is over the randomness of $D, V$ and possible algorithm randomness.*

*Proof.* This follows from Theorem 1 in Carmon et al. (2019b) (or equivalently Theorem 6 in Schmidt et al. (2018)). The only difference of our setting from their setting is that we additionally have unlabeled data $U$ for the algorithm. Since the attacker can provide $x' = x$, the problem reduces to a problem with at most $n + 1$ data points in their setting, and thus the statement (1) follows. □

**Transductive learning algorithms** $(\mathcal{T}, \Gamma)$: To prove the statement (2), we give concrete algorithms learning algorithms that achieves small test error on $x'$.

**High-level structure of the learning algorithms.** At the high level, the learning algorithms work as follows: At the training time we use part of the training data (denoted as $D_2$ to train a pretrained model $\bar{\theta}$), and part of the training data (denoted as $D_1$, is reserved to test-time adaptation). Then, at the test time, upon receiving $U$, we use $U$ to tune $\bar{\theta}$, and get two large-margin classifiers, $\bar{\theta}_+$ and $\bar{\theta}_-$, which classify $x'$ as $+1$ and $-1$, respectively. Finally, we check these two large margin classifiers on $D_1$ (that's where $D_1$ is used), and the one that generates smaller error wins and we classify $x'$ into the winner class.

**Detailed description.** More specifically, the learning algorithms $(\mathcal{T}, \Gamma)$ work as follows:

1. **Before game starts**. Let $m' = Kn_0, m = 10n_0$. We split the training set $D$ into two subsets: $D_1 := \{(x_i, y_i)\}_{i=1}^{m'}$ and $D_2 := \{(x_{m'+i}, y_{m'+i})\}_{i=1}^{m}$. $D_2$ will be used to train a pretrained model at the training time, and $D_1$ will be used at the test time for adaptation.
2. **Training time**. $\mathcal{T}$ uses the second part $D_2$ to compute a pretrained model, that is, a parameter vector:

$$\hat{\theta}_m = \frac{1}{m}\sum_{i=1}^{m} y_{m'+i}x_{m'+i}, \quad \bar{\theta} = \frac{\hat{\theta}_m}{\|\hat{\theta}_m\|_2}. \tag{6}$$

3. **Test time**. On input $U$, $\Gamma$ uses $D_1$ and $U$ to perform adaptation. At the high level, it adapts the pre-trained $\bar{\theta}$ along the direction of $x'$, such that it also has a large margin on $x'$, and also it makes correct predictions on $D_1$ with large margins. More specifically:

(a) First, $\Gamma$ constructs two classifiers, $\theta_+$ and $\theta_-$, such that $\theta_+$ classifies $x'$ to be $+1$ with a large margin, and $\theta_-$ classifies $x'$ to be $-1$ with a large margin. Specifically:

$$\bar{x}' := x'/\|x'\|_2, \qquad\qquad \gamma := \|x'\|_2/2, \qquad\qquad\qquad (7)$$

$$\eta_+ := \frac{\gamma - (\bar{\theta})^\top x'}{\|x'\|_2}, \qquad \theta_+ = \bar{\theta} + \eta_+ \bar{x}', \qquad \bar{\theta}_+ = \theta_+/\|\theta_+\|_2, \quad (8)$$

$$\eta_- := \frac{-\gamma - (\bar{\theta})^\top x'}{\|x'\|_2}, \qquad \theta_- = \bar{\theta} + \eta_- \bar{x}', \qquad \bar{\theta}_- = \theta_-/\|\theta_-\|_2. \quad (9)$$

where $\theta_+$ and $\theta_-$ are viewed as the parameter vectors for linear classifiers. Note that $\theta_+$ is constructed such that $\theta_+^\top x'/\|x'\|_2 = \gamma/\|x'\|_2 = 1/2$, and $\theta_-$ is such that $\theta_-^\top x'/\|x'\|_2 = -\gamma/\|x'\|_2 = -1/2$.

(b) Finally, $\Gamma$ checks their large margin errors on $D_1$. Formally, let

$$t := \sigma \left( \sqrt{\frac{n_0}{d}} + \frac{n_0}{m} \right)^{-1/2}, \qquad\qquad (10)$$

$$\mathrm{err}_t(\theta) := \mathbb{E}_{(x,y)} \mathbb{I}[y\theta^\top x \leq t], \qquad\qquad (11)$$

$$\widehat{\mathrm{err}_t}(\theta) := \frac{1}{m'} \sum_{i=1}^{m'} \mathbb{I}[y_i\theta^\top x_i \leq t]. \qquad\qquad (12)$$

If $\widehat{\mathrm{err}_t}(\bar{\theta}_+) \leq \widehat{\mathrm{err}_t}(\bar{\theta}_-)$, then $\Gamma$ sets $\widetilde{F}(x) := \mathrm{sgn}(\bar{\theta}_+^\top x)$ and classifies $x'$ to $+1$; otherwise, it sets $\widetilde{F}(x) := \mathrm{sgn}(\bar{\theta}_-^\top x)$ and classifies $x'$ to $-1$.

**Lemma 2 (Part (2)).** *In the maximin threat model, there is an absolute constant $c > 0$, such that for the $\mathcal{T}$ and $\Gamma$ described above, the adapted model $\widetilde{F} = \Gamma(\mathcal{T}(D), D, U)$ has a small error:*

$$\mathbb{E}\left\{ L(\widetilde{F}, \widetilde{V}) \right\} \leq e^{-cK}. \qquad\qquad (13)$$

*Proof.* Now, we have specified the algorithms and are ready to prove that w.h.p. $\widetilde{F}(x')$ is the correct label $y$. By Lemma 3, $y(\mathrm{err}_t(\bar{\theta}_-) - \mathrm{err}_t(\bar{\theta}_+)) \geq \frac{c_4}{\sqrt{n_0}}$ with probability $\geq 1 - e^{-c_4 K}$. Then by the Hoeffding's inequality, $D_1$ is sufficiently large to ensure $y(\widehat{\mathrm{err}_t}(\bar{\theta}_+) - \widehat{\mathrm{err}_t}(\bar{\theta}_-)) > 0$ with probability $\geq 1 - 2e^{-c_4^2 K/2}$. This proves the statement (2). $\qquad\square$

**Lemma 3.** *There exists absolute constants $c_4 > 0$ such that with probability $\geq 1 - e^{-c_4 K}$,*

$$y(\mathrm{err}_t(\bar{\theta}_-) - \mathrm{err}_t(\bar{\theta}_+)) \geq \frac{c_4}{\sqrt{n_0}}. \qquad\qquad (14)$$

## F.1 PROOF OF LEMMA 3

Without loss of generality, assume $y = +1$. The proof for $y = -1$ follows the same argument.

Note that

$$\mathrm{err}_t(\theta) = \mathbb{E}_{(x,y)} \mathbb{I}[y\theta^\top x \leq t] \qquad\qquad (15)$$

$$= \mathbb{P}\left( \mathcal{N}(\mu^\top \theta, \sigma^2 \|\theta\|_2^2) \leq t \right) \qquad\qquad (16)$$

$$= Q\left( \frac{\mu^\top \theta - t}{\sigma \|\theta\|_2} \right), \qquad\qquad (17)$$

where

$$Q(x) := \frac{1}{\sqrt{2\pi}} \int_x^{+\infty} e^{-t^2/2} dt. \qquad\qquad (18)$$

First, consider $\bar{\theta}$.

$$\mathrm{err}_t(\bar{\theta}) = Q\left( \frac{\mu^\top \bar{\theta} - t}{\sigma \|\bar{\theta}\|_2} \right) = Q(s), \text{ where } s := \frac{\mu^\top \bar{\theta} - t}{\sigma}. \qquad (19)$$

By Lemma 4, we have with probability $\geq 1 - e^{-c_2(d/n_0)^{1/4} \min\{m,(d/n_0)^{1/4}\}}$,

$$\frac{\mu^\top \bar{\theta}}{\sigma \|\bar{\theta}\|_2} \leq \left( \sqrt{\frac{n_0}{d}} + \frac{n_0}{m} \right)^{-1/2} \left( 1 + c_1 \left( \frac{n_0}{d} \right)^{1/8} \right), \tag{20}$$

$$\frac{\mu^\top \bar{\theta}}{\sigma \|\bar{\theta}\|_2} \geq \left( \sqrt{\frac{n_0}{d}} + \frac{n_0}{m} \right)^{-1/2} \left( 1 - c_1 \left( \frac{n_0}{d} \right)^{1/8} \right), \tag{21}$$

which gives

$$s = \frac{\mu^\top \bar{\theta} - t}{\sigma \|\bar{\theta}\|_2} \leq c_1 \left( \frac{n_0}{d} \right)^{1/8} \left( \sqrt{\frac{n_0}{d}} + \frac{n_0}{m} \right)^{-1/2}, \tag{22}$$

$$s = \frac{\mu^\top \bar{\theta} - t}{\sigma \|\bar{\theta}\|_2} \geq -c_1 \left( \frac{n_0}{d} \right)^{1/8} \left( \sqrt{\frac{n_0}{d}} + \frac{n_0}{m} \right)^{-1/2}. \tag{23}$$

Since $m = 10n_0$ and $d \gg n_0$, we have

$$|s| = \left| \frac{\mu^\top \bar{\theta} - t}{\sigma \|\bar{\theta}\|_2} \right| \leq 1. \tag{24}$$

Next, we have

$$\text{err}_t(\theta_+) = Q \left( \frac{\mu^\top \bar{\theta}_+ - t}{\sigma \|\bar{\theta}_+\|_2} \right) = Q(s_+), \text{ where } s_+ := \frac{\mu^\top \bar{\theta}_+ - t}{\sigma}, \tag{25}$$

$$\text{err}_t(\theta_-) = Q \left( \frac{\mu^\top \bar{\theta}_- - t}{\sigma \|\bar{\theta}_-\|_2} \right) = Q(s_-), \text{ where } s_- := \frac{\mu^\top \bar{\theta}_- - t}{\sigma}. \tag{26}$$

We now check the sizes of $s_+$ and $s_-$.

$$s_+ - s = \frac{\mu^\top \bar{\theta}_+ - t}{\sigma} - \frac{\mu^\top \bar{\theta} - t}{\sigma} \tag{27}$$

$$= \frac{\mu^\top \bar{\theta}_+ - \mu^\top \bar{\theta}}{\sigma} \tag{28}$$

$$= \frac{1}{\sigma \|\theta_+\|_2} \left( (1 - \|\theta_+\|_2) \mu^\top \bar{\theta} + \eta_+ \mu^\top \bar{x}' \right). \tag{29}$$

Then by definition and bounds in Claim 1,

$$|s_+ - s| \leq \frac{2}{n_0} + 40 \leq 42. \tag{30}$$

Since $|s|$ is bounded by 1, we know $|s_+|$ is also bounded by 43. Similarly, $|s_- - s|$ and thus $|s_-|$ are also bounded by some constants. Furthermore,

$$s_+ - s_- = \frac{1}{\sigma} \left( \mu^\top \bar{\theta}_+ - \mu^\top \bar{\theta}_- \right) \tag{31}$$

$$= \frac{1}{\sigma} \left( \frac{\mu^\top \bar{\theta} + \eta_+ \mu^\top \bar{x}'}{\|\theta_+\|_2} - \frac{\mu^\top \bar{\theta} + \eta_- \mu^\top \bar{x}'}{\|\theta_-\|_2} \right). \tag{32}$$

By Claim 2, we have $\|\theta_-\|_2 = \|\theta_+\|_2$. So

$$s_+ - s_- = \frac{1}{\sigma \|\theta_+\|_2} \left( \eta_+ \mu^\top \bar{x}' - \eta_- \mu^\top \bar{x}' \right) \tag{33}$$

$$= \frac{1}{\sigma \|\theta_+\|_2} \left( \eta_+ - \eta_- \right) \mu^\top \bar{x}' \tag{34}$$

$$= \frac{1}{\sigma \|\theta_+\|_2} \mu^\top \bar{x}' \tag{35}$$

$$\geq \frac{\sqrt{d}}{4\sigma^2} \tag{36}$$

$$= \frac{1}{4\sqrt{n_0}}. \tag{37}$$

Now we are ready to bound the error difference:

$$\text{err}_t(\bar{\theta}_-) - \text{err}_t(\bar{\theta}_+) = Q(s_-) - Q(s_+) \tag{38}$$

$$= \frac{1}{\sqrt{2\pi}} \int_{s_-}^{s_+} e^{-t^2/2} dt \tag{39}$$

$$\geq \frac{1}{\sqrt{2\pi}} (s_- - s_+) \times \min\{e^{-s_-^2/2}, e^{-s_+^2/2}\} \tag{40}$$

$$\geq \frac{c_4}{\sqrt{n_0}} \tag{41}$$

for some absolute constant $c_4 > 0$.

## F.2 Tools

**Claim 1.** *There exists a absolute constant $c_3 > 0$, such that with probability $\geq 1 - e^{-c_3 K}$,*

$$\sigma\sqrt{d}/4 \leq \|x'\|_2 \leq 2\sigma\sqrt{d}, \tag{42}$$

$$\frac{1}{2}\sigma \leq \frac{1}{4}\sigma\sqrt{\frac{m}{n_0}} \leq \bar{\theta}^\top \mu \leq 2\sigma\sqrt{\frac{m}{n_0}} \leq 10\sigma, \tag{43}$$

$$-\epsilon\sqrt{d}/2 \leq \bar{\theta}^\top x' \leq 2\epsilon\sqrt{d}, \tag{44}$$

$$d/2 \leq \mu^\top x' \leq 3d/2, \tag{45}$$

$$\frac{1}{2} - \frac{8\epsilon}{\sigma} \leq \eta_+ \leq \frac{1}{2} + \frac{8\epsilon}{\sigma}, \tag{46}$$

$$-\frac{1}{2} - \frac{8\epsilon}{\sigma} \leq \eta_- \leq -\frac{1}{2} + \frac{8\epsilon}{\sigma}. \tag{47}$$

*Proof.* First, since $x' = \mu + \sigma\zeta + \nu$ for $\zeta \sim \mathcal{N}(0, I)$, with probability $\geq 1 - e^{-c'd}$ for an absolute constant $c' > 0$, we have:

$$\sqrt{d}/2 \leq \|\zeta\|_2 \leq 3\sqrt{d}/2, \tag{48}$$

$$\|x'\|_2 \geq \sigma\sqrt{d}/2 - \|\mu\|_2 - \|\nu\|_2 \geq \sigma\sqrt{d}/4, \tag{49}$$

$$\|x'\|_2 \leq \sigma 3\sqrt{d}/2 + \|\mu\|_2 + \|\nu\|_2 \leq 2\sigma\sqrt{d}. \tag{50}$$

By Lemma 4, with probability $\geq 1 - e^{-c_2 K}$,

$$\bar{\theta}^\top \mu \leq 2\sigma \left( \sqrt{\frac{n_0}{d}} + \frac{n_0}{m} \right)^{-1/2} \leq 2\sigma\sqrt{\frac{m}{n_0}}, \tag{51}$$

$$\bar{\theta}^\top \mu \geq \frac{1}{2}\sigma \left( \sqrt{\frac{n_0}{d}} + \frac{n_0}{m} \right)^{-1/2} \geq \frac{\sigma}{4}\sqrt{\frac{m}{n_0}}. \tag{52}$$

Also, with probability $1 - e^{-c'K}$,

$$|\bar{\theta}^\top \zeta| \leq 2K\sigma. \tag{53}$$

Finally,

$$|\bar{\theta}^\top \nu| \leq \|\bar{\theta}\|_1 \|\nu\|_\infty \leq \epsilon\sqrt{d}. \tag{54}$$

Then

$$\bar{\theta}^\top x' = \bar{\theta}^\top (\mu + \sigma\zeta + \nu) \tag{55}$$

$$\leq |\bar{\theta}^\top \mu| + \sigma|\bar{\theta}^\top \zeta| + |\bar{\theta}^\top \nu| \tag{56}$$

$$\leq 2\sigma\sqrt{\frac{m}{n_0}} + 2K\sigma + \epsilon\sqrt{d} \tag{57}$$

$$\leq 2\epsilon\sqrt{d}. \tag{58}$$

and

$$\bar{\theta}^\top x' = \bar{\theta}^\top (\mu + \sigma\zeta + \nu) \tag{59}$$

$$\geq \sigma/2 - K\sigma - \epsilon\sqrt{d} \tag{60}$$

$$\geq -\epsilon\sqrt{d}/2. \tag{61}$$

For $\mu^\top x'$, we have with probability $\geq 1 - e^{-c'K}$,

$$\mu^\top x' = \mu^\top (\mu + \sigma\zeta + \nu) \tag{62}$$

$$\mu^\top x' \leq \|\mu\|_2^2 + 2K\sigma\|\mu\|_2 + \epsilon\|\mu\|_2\sqrt{d} \leq 3d/2, \tag{63}$$

$$\mu^\top x' \geq \|\mu\|_2^2 - 2K\sigma\|\mu\|_2 - \epsilon\|\mu\|_2\sqrt{d} \geq d/2. \tag{64}$$

By definition:

$$\eta_+ = \frac{1}{2} - \bar{\theta}^\top x'/\|x'\|_2, \tag{65}$$

so

$$\frac{1}{2} - 8\epsilon/\sigma \leq \eta_+ \leq \frac{1}{2} + 8\epsilon/\sigma. \tag{66}$$

Similarly,

$$-\frac{1}{2} - 8\epsilon/\sigma \leq \eta_- \leq -\frac{1}{2} + 8\epsilon/\sigma. \tag{67}$$

$\square$

**Claim 2.**

$$\|\theta_+\|_2 = \|\theta_-\|_2. \tag{68}$$

*Proof.* We have by definition:

$$\|\theta_-\|_2^2 = \|\bar{\theta} + \eta_-\bar{x}'\|_2^2 \tag{69}$$

$$= 1 + \eta_-^2 + 2\eta_-\bar{\theta}^\top\bar{x}', \tag{70}$$

$$\|\theta_+\|_2^2 = \|\bar{\theta} + \eta_+\bar{x}'\|_2^2 \tag{71}$$

$$= 1 + \eta_+^2 + 2\eta_+\bar{\theta}^\top\bar{x}'. \tag{72}$$

Then

$$\|\theta_-\|_2^2 - \|\theta_+\|_2^2 = \eta_-^2 + 2\eta_-\bar{\theta}^\top\bar{x}' - \eta_+^2 - 2\eta_+\bar{\theta}^\top\bar{x}' \tag{73}$$

$$= (\eta_- - \eta_+)(\eta_- + \eta_+) + 2\bar{\theta}^\top\bar{x}'(\eta_- - \eta_+) \tag{74}$$

$$= (\eta_- - \eta_+)[(\eta_- + \eta_+) + 2\bar{\theta}^\top\bar{x}'] \tag{75}$$

$$= (\eta_- - \eta_+)[-2\bar{\theta}^\top x'/\|x'\|_2 + 2\bar{\theta}^\top\bar{x}'] \tag{76}$$

$$= 0. \tag{77}$$

This completes the proof. $\square$

**Lemma 4** (Paraphrase of Lemma 1 in Carmon et al. (2019b)). *Let* $\hat{\theta}_m = \frac{1}{m}\sum_{i=1}^m y_i x_i$. *There exist absolute constants* $c_0, c_1, c_2$ *such that under parameter setting* (1) *and* $d/n_0 > c_0$,

$$\frac{\sigma^2\|\hat{\theta}_m\|_2^2}{(\mu^\top\hat{\theta}_m)^2} \geq \left(\sqrt{\frac{n_0}{d}} + \frac{n_0}{m}\right)\left(1 - c_1\left(\frac{n_0}{d}\right)^{1/8}\right), \tag{78}$$

$$\frac{\sigma^2\|\hat{\theta}_m\|_2^2}{(\mu^\top\hat{\theta}_m)^2} \leq \left(\sqrt{\frac{n_0}{d}} + \frac{n_0}{m}\right)\left(1 + c_1\left(\frac{n_0}{d}\right)^{1/8}\right), \tag{79}$$

*with probability* $\geq 1 - e^{-c_2(d/n_0)^{1/4}\min\{m,(d/n_0)^{1/4}\}}$.

