# OpenReview forum: "Test-Time Adaptation and Adversarial Robustness"
_ICLR.cc/2021/Conference — Reject_

### Official Review · AnonReviewer4 · 2020-10-26
**Review of "Test-Time Adaptation and Adversarial Robustness"**

**Rating:** 5
**Confidence:** 3

**Review:**

### Contributions ###
* The authors formalize test-time adaptation as a maximin threat model and contrast it with the "threat model for classic adversarial robustness" and the "adversarial semi-supervised minimax threat model".
* The paper studies test-time adaptation via DANN and its robustness in the maximin and minimax threat model. DANN has larger robustness against the proposed attacks in the maximin threat model than in minimax, providing evidence that maximin  is a setting that is beneficial for the defender. However, it remains unclear if this would also be the case against stronger attackers.

### Significance ###

Increasing robustness against adversarial robustness is a topic both relevant to basic research and also more applied research. Test-time adaptation is currently one of the most promising directions and this paper is thus very timely. Its formalization of a threat model for this domain can guide future research in this direction.
That being said, the main method investigated (DANN for test-time adaptation) seems less practical and significant since it requires large test batch sizes and also huge computational resources at inference time.

### Originality ###

The paper has only limited novelty: while the formalization of test-time adaptation as maximin threat model is very thorough, it is also mostly straightforward. Also studying DANN as a test-time adaptation is an incremental idea once one assumes that inference is conducted on large batches. Nevertheless, I think papers like this that consolidate recent research directions into a coherent framework are valuable.

### Clarity ###

Generally, the introduction and the part outlining the threat model are written very clearly and I enjoyed the thorough formalization. A few details could be improved (even though this is a bit nitpicking):
* the introduction is very technical; it could be more high level and shorter and leaving the more technical parts to Section 2.
* Threat models could mention that white-box knowledge of the attacker is assumed
* it is not really clear why test inputs are assumed to be labeled in the threat models

The experimental part is less clear unfortunately:
* since DANN is a central method in the experiments, it should be briefly summarized to make the paper self-contained
* it is not really clear to me how specifically DANN is trained here (what is the objective?). Specifically, is the loss on the labeled data based on standard training or adversarial training?
* generally, all training details (optimizer, learning rate, batch size, number of epochs etc.) are missing. the experiments cannot be reproduced in the current form.

### Quality ###

I have some serious doubts about the quality of the empirical evaluation.
As the authors state "While we are not able to prove the existence of a defender solution
that separates maximin and minimax threat models, we design and implement experiments to
provide evidence that such a strategy may actually exist." Thus, one of the main hypothesis of the paper (maximin being a strictly weaker threat model than minimax) depends on the design of experiments. My main concern here is that the attackers used are not actually strong:
* A transfer attack should clearly only be a baseline that should get surpassed by stronger adaptive attackers
* However, the fixed point attack seems to be weaker than the transfer attack. In particular its effectiveness in the homogeneous setting for CIFAR-10 reduces with the number of iterations k. This does not seem to be a reliable way of evaluating DANN's maximin robustness
* I would imagine a strong adaptive attack could be built by generating adversarial inputs  with the objective of minimizing the loss of DANN's domain label classifier. This would minimize the impact of test-time adaptation via DANN (because both domains would have already very similar representations). In any case, more effort should be spend on designing a strong attack against a DANN-based test-time adaptation.

I think it is also confusing to add evaluations in the inhomogeneous setting. Clearly, DANN gives strong results here because dealing with domain shift is what it was designed for. However, the paper is on adversarial robustness and confounding this with domain shift is misleading (my suspicion is it was mainly added because DANN clearly outperforms adversarial training on the source data in this setting; to proof me wrong, the authors should add a better motivation for this setting or remove it).

### Recommendation ###

To summarize, I think the paper provides a strong formalization of a threat model for test-time adaptation. However, it does not introduce novel approaches, lacks clarity in experimental details, and its empirical results are less trustworthy given that the proposed attacks are rather weak. I thus recommend rejecting the paper in its current form. I encourage the authors to address the current shortcomings; the paper clearly has potential in general.

### Final Recommendation after Author Response ###
I would like the authors for the very active and productive rebuttal period.  Actually, the current version has significantly deviated from the initially submitted version. I have read most of the paper a second time to take all changes into account. The authors considerably improved the paper and have addressed some of my concerns. I increase my score to 5. The reason for not increasing the score to 6 or 7 is mainly that clarity of the paper is lacking. To give two examples:
 * Presumably because of the many changes during the rebuttal, the organization, structure and the quality of the manuscript is not always sufficient: to name one of several examples, the core Theorem 1 comes without any reference to its proof or any proof sketch in the main paper (there exists a proof in the appendix, finding of which required scrolling through the entire appendix).
 * As stated in my initial review, since DANN is a central method in the experiments, it should be briefly summarized to make the paper self-contained and also the specific training
I think the core issue with the submission is that there is simply too much content for one paper:
 * formalizing 3 threat models
 * a proof of separation of maximin and minimax
 * empirical evaluation  on two datasets (MNIST, CIFAR10), three attacks (transfer, two adaptive attacks), three defenses (DANN, AdvS, TTT), and two settings (homogenous, inhomogeneous)
Because of the page limit, a lot of details have been moved into the appendix, making the paper difficult to read. Moreover, even taking the appendix into account, details remain unclear such as how DANN was trained. Moreover, I do not find it convincing to move related work to the appendix; relating the current work to other work should be an integral part of the main paper.

My recommendation for the authors would be to strengthen the focus of the paper: I think the inhomogeneous setting does not contribute much and could be removed. Also I don't see much value in MNIST and the weak transfer attacks. Moreover, the parts on preliminaries and threat models is too long for a conference paper and could be shortened.
I think if focus were improved and the main document became more self-contained, the quality of the work would be considerably improved. For the time being, I see the submission still marginally below the acceptance threshold.

---

> ### Author Response · Authors · 2020-11-10
> **Appreciate for recognizing novelty, but we disagree that the approach is not novel**
>
> We sincerely appreciate the reviewer for correctly recognizing the novelty for formulating a threat model for test-time adaptation.
> We want to, however, rebut first on some criticisms that are critical to understanding our contributions:
>
> 1. First, to set up the context correctly, we want to point out that we are using the original DANN objective (designed for unsupervised domain adaptation) as defined in: https://arxiv.org/abs/1505.07818 (see equation (9) of the paper). For an intuitive understanding, one can also refer to this link
> https://ameroyer.github.io/reading-notes/domain%20adaptation/2019/05/23/domain_adversarial_training_of_neural_networks.html
>
> 2. We want to highlight that this objective is **attack agnostic**, namely in the objective it never explicitly leverages the information of how the adversarially perturbed data are generated (i.e., it does not leverage the attack type in the objective). Specifically, this  answers your question of *"it is not really clear to me how specifically DANN is trained here (what is the objective?). Specifically, is the loss on the labeled data based on standard training or adversarial training?"*. No, the training on the source data is not adv training at all: If it was the case, this paper would lose all its meaning. And the fact that DANN can provide adversarial robustness is exactly why transductive adversarial learning can be surprising and interesting.
>
> 3. We believe that the use of DANN is novel. Before we introduced our threat model, it is not clear at all why one should even consider DANN (an algorithm for unsupervised domain adaptation) as an algorithm for **adversarial** robustness. Our research builds a bridge to connect two seemingly entirely different research directions, which to us is a clear novelty.
>
> 4. Finally, we also want to rebut in this thread about **"threat model being straightforward"**. We understand where this comes from, but this is an issue of the nature of "searching for a solution vs. verifying a solution". As a conceptual paper, being able to articulate a threat model that is natural and clean is an advantage, rather than a disadvantage. To this end, we want to emphasize that the recent beautiful theory work by Goldwasser et al. (spotlight in neurips 2020, https://arxiv.org/abs/2007.05145; Beyond Perturbations: Learning Guarantees with Arbitrary Adversarial Test Examples) have **studied a threat model similar in spirit to ours (see their Definition 1.1 and Section 4.2)**. However, while their threat model is more amenable to theoretical analysis, we view ours as much more practical and relevant to deep learning (specifically, their formulation is designed to be analyzable for bounded VC dimension concept class, which may not be practical in the deep learning setting). Our choices of formalization have made it **possible** to connect to deep learning, and demonstrate that unsupervised domain adaptation can benefit adversarial robustness, which to us is very nontrivial.
>
> For your other constructive comments, we will post a separate thread to address those concerns.

---

> > ### Author Response · Authors · 2020-11-10
> > **Clarifications for evaluating on the inhomogeneous setting**
> >
> > The reviewer challenged about the motivations for evaluating in the inhomogeneous setting. While we agree that DANN clearly has
> > an advantage, the motivation of this setting is indeed well founded. Below we list a few related points:
> >
> > 1. To start with, test-time adaptation (such as TTT for example) has been very commonly applied in the "different domain setting."
> > For example, TTT considered a setting where we try to adapt from CIFAR10 to CIFAR10 corrupted (the CIFAR10-c dataset is designed
> > in Benchmarking Neural Network Robustness to Common Corruptions and Perturbations (ICLR 2019), by Dan Hendrycks and Thomas Dietterich). Therefore, it is a very natural question, if we want to study adversarial robustness of test-time adaption after all, whether we can provide adversarial robustness when training set is CIFAR10, and test set is CIFAR10-c.
> >
> > 2. To this end, we note that several of the previous works (we cited) have started to investigate robustness issues in domain adaptation (such as CIFAR10->CIFAR10-c). We believe that generalizing those to **adversarial** robustness is not only natural, but also, highlights the surprising power of transductive learning (which gets adversarial robustness even in the inhomogeneous setting).
> >
> > 3. As we have mentioned earlier, the recent theory work Goldwasser et al. (https://arxiv.org/abs/2007.05145) investigated a surprising setting where they showed that, for bounded VC concept class, **transductive adversarial learning** can provide robustness even with **arbitrary** adversarial test time examples. This includes domain shift as a particular special case.
> >
> > 4. Following (3), of course, it is hard to imagine that, for deep learning, such guarantees hold (which is far from having bounded VC dimensions). Nevertheless, our work has shown that DANN can provide robustness at least in the familiar setting of unsupervised domain adaptation (and further the test examples can be **adversarially perturbed**). To this end, as we have mentioned, our work naturally complemented Goldwasser et al.'s work in the setting of deep learning.
> >
> > We hope that these points clarify the motivations for evaluating inhomogeneous settings. It is critical and natural for test-time adaptation, and highlights the power of transductive adversarial learning.

---

### Official Review · AnonReviewer2 · 2020-10-27
**A potentially interesting new research direction, but many questions need to be answered beforehand.**

**Rating:** 4
**Confidence:** 4

**Review:**

This paper explores a new paradigm referred to as "adversarial robustness of test-time adaptation". The authors propose a new threat model called a "maximin game" instead of the widely-accepted "minimax game". In this new setup, the threat model becomes weaker as it first presents the unlabelled adversarial examples to the defender to get prepared. The authors also run some experiments showing that DANN seems to be a preferred method in this new setup. The paper is in general interesting with many novel definitions and discussions, but I do not see how this paper can benefit the community other than a set of new research directions with preliminary results.

1. First of all, I consider the writings of the introduction not professional: the introduction is full of notations, definitions, and even datasets and results that are only elucidated later in the paper. This writing style posted a great challenge to understand and appreciate this paper.

2. The paper interestingly introduces a set of new ideas/definitions. However, I wonder how these new research directions can benefit the community or industry. The authors explained that "this question is also of practical importance since these 'new' solutions may possess desirable properties that good solutions in the minimax threat model may lack." and then the authors mentioned that "one such property is that the defender solution is attack agnostic". However, it seems the rest of this paper does not correspond to these remarks, e.g.,
 * The experiments (in the main paper body) are within $\ell_\infty$ norm, so it seems there is no evidence about attack agnostic.
 * I did not find related discussions of other "desirable properties" later in this paper.
 * The authors also mentioned theoretical interest, yet I don't think, with the current contents, this paper can be appreciated as a theoretical paper.
Therefore, I suggest the authors further clarify their contributions, with emphasis on how their contributions can benefit the community and the industry. The current presentation leaves the impression that the paper is a set of new definitions without real-world implications.

3. The paper also says "This threat model enables us to study whether a large test set can benefit a defender for adversarial robustness", yet I do not see any experiments (in the main paper) that correspond to this discussion. The appendix seems lacking this discussion either. Similarly, this kind of argument or claims without validations makes the paper read like an arbitrary set of definitions.

4. Following the "attack agnostic" argument above, Definition 2 suggests that the solution is also attack specific, as the defender needs to update the model according to the attacked samples. Although this update happens at test time, it happens before the evaluation.

5. The experiments are only about DANN, and seemingly leaving the details of AdvS unexplained. Is the AdvS the same architecture but without the domain component? Then why not also report performances on S? Without these details, it's impossible to evaluate and appreciate the experimental results.

6. While it seems the authors' focus is data-oblivious test-time adaptation, which, as I understand, does not allow the usage of labeling training data. Yet a major focus of the paper is DANN, which allows such usage. I'm aware that the authors have mentioned the differences in multiple places, but if the main focus is DANN, I suggest the authors rephrase the paper to be about the setup DANN builds upon, instead of targeting a broader domain but only supporting the claims with experiments limited to a specific setup.

---

> ### Author Response · Authors · 2020-11-10
> **Quick replies on some critical misread**
>
> Thanks for the review. We found that the comments have some significant misread, which we want to point out right away:
>
> 1.  Quote: *"and then the authors mentioned that "one such property is that the defender solution is attack agnostic". However, it seems the rest of this paper does not correspond to these remarks, e.g., The experiments (in the main paper body) are within  norm, so it seems there is no evidence about attack agnostic."*
>
> This is a misread. We have shown that DANN works as a test-time defense mechanism, and DANN is agnostic to the $\ell_\infty$ attacks: This is because that we are using the original DANN algorithm,  which is not specifically designed for the $\ell_\infty $ attacks. In the introduction we have specifically mentioned: **"This is somewhat surprising as DANN is also agnostic to $\ell_\infty$ attacks"**. On pp. 6 in experiments we have also specifically mentioned, quote: **"(B) DANN is however not designed for adversarial robustness. Thus is it a very interesting question whether DANN can provide test-time
> adversarial robustness against attacks (e.g. norm-based attacks) that it is not specifically designed
> for"** Please refer to point 3 below for more discussion about "attack agnostic".
>
> 2. Quote: *"whether a large test set can benefit a defender for adversarial robustness", yet I do not see any experiments (in the main paper) that correspond to this discussion"*
>
> Please refer to Figure 1 in the experiments: We have explicitly experimented with decreasing test size, and the robustness decreases.
>
> 3. Quote:  *"Following the "attack agnostic" argument above, Definition 2 suggests that the solution is also attack specific, as the defender needs to update the model according to the attacked samples."*
>
> We are afraid that there is a misunderstanding of what do we mean by "attack agnostic". It means that the defense mechanism
> (the algorithm) does not leverage explicitly the attack type. For example, while DANN algorithm leverages $U$, the **DANN algorithm itself never explicitly uses the fact that the adversarial data is from $\ell_\infty$ attacks**. Also, in Section D.3 we have also shown that DANN algorithm can provide robustness for $\ell_2$ attacks as well, which further corroborates the "attack agnostic point". We feel that this is critical to the understanding, so we want to point out as early as possible.
>
> 4. Quote: *"While it seems the authors' focus is data-oblivious test-time adaptation, which, as I understand,..."*
>
> We understand where this came from, but an important component of this paper is that we show that the current data-oblivious test-time adaptations (e.g. Test-Time Training) fails to have any robustness. Because the experiments for that purpose are so easy to perform, we condense them into Section 5 (but they are important).

---

> > ### Comment · AnonReviewer2 · 2020-11-11
> > **About "attack agnostic"**
> >
> > Thanks for the quick response, by the misalignment between the claim "attack agnostic" and the results are shown in the experiment, I mainly refer to the fact that, no matter what, your method need the model to be exposed to a specific type of attacks: although this attack happens at test time, it happens before evaluation time. Thus, the method is not "attack agnostic". Whether you can train DANN with the knowledge of attack or not is barely relevant, because I can hardly imagine this part of the model plays a critical role here.
> >
> > The authors also seem to argue that they can define "attack agnostic" differently, which I recommend the authors not to, because those phrasing can easily mislead readers.
> >
> > Regarding the data size problems, I tend to not consider experiments with a specific dataset with three specific fixed configurations (Figure 1) that can contribute to a conclusion. If the authors want to draw such a conclusion, please conduct more comprehensive experiments, or please deemphasize the related texts, describing relevant results as something additional, instead of a benefit of the method.
> >
> > Finally, a friendly reminder that the authors have not addressed all the concerns raised.

---

> > > ### Author Response · Authors · 2020-11-11
> > > **We now understand where you are coming from, but that does not make sense**
> > >
> > > First, **any** nontrivial transductive defense will necessarily use the information of $U$, so according to your definition, none of the transductive defense could possibly be "attack agnostic". So at least in the case of transductive defenses,  if one wants nontrivial meaning after all, one has to define "attack agnostic" different than your proposal.
> > >
> > > Second, our definition of "attack agnostic" follows the same proposal by Goodfellow's paper:  Defense against the dark arts: An overview of adversarial example security research and future research directions. (https://arxiv.org/abs/1806.04169), where he mentioned, on pp.30 (which we write in the intro), quote: *"the solution is not to directly optimize the performance measure for a
> > > particular type of perturbation"*. Here he means that the existing adversarial training are all optimization explicitly to train for an attack type. It is from this we define our notion of "attack agnostic", and we notice that DANN is exactly attack agnostic, because the optimization objective and algorithm never explicitly optimize for a particular attack type, and our experiments show that it provides robustness for both $\ell_\infty$ and $\ell_2$ attacks. This is an interesting, and somewhat surprising, connection.
> > >
> > > Third, you were saying that we don't have experiments on the effect of batch size, and we just want to let you know there is. Revising
> > > those experiments is a different matter. We will revise on experiments and reply.
> > >
> > > Finally, thanks for the reminder. We are aware that we have not addressed all concerns. We are still working on it. Thanks.

---

> > > > ### Comment · AnonReviewer2 · 2020-11-11
> > > > **Follow up in "attack agnositic"**
> > > >
> > > > Exactly, none successful defense can be "attack agnostic".
> > > >
> > > > If "the solution is not to directly optimize the performance measure for a particular type of perturbation", then I'm happy to call "attack agnostic", but you didn't follow that: the model is exposed to the attack and updates its parameters before evaluation.
> > > >
> > > > If you want to convince me that your method is "attack-agnostic", at the test time, when the attacker plays a role, please use something as simple as Gaussian noises, and then after the defender updates the parameters and when it comes to the evaluation, use whatever attack you prefer $\ell_\infty$ or $\ell_2$. Please report some preliminary results on this (if you want to convince me your method is "attack agnostic").
> > > >
> > > > However, please notice this is only a first step, even if you can show the model is "robust", I will suggest you start to follow "On Evaluating Adversarial Robustness", as other reviewers suggested, to test whether the model is actually robust.

---

> > > > > ### Author Response · Authors · 2020-11-11
> > > > > **You are confused about "evaluation" in the threat model**
> > > > >
> > > > > In the threat model, the attacker can only perform the steps marked as attacker, and the steps that are not marked belong to the "referee". Specifically, the last step of evaluating accuracy belongs to the referee. Namely in the threat model, we ask the referee to judge the accuracy of the adapted model -- because only with that we can formally analyze the performance of the method.
> > > > >
> > > > > Things like these are very common in the formulation of threat models (e.g., theoretical studies of cryptography).
> > > > >
> > > > > We hope this can clarify for you.

---

> > > > > > ### Comment · AnonReviewer2 · 2020-11-11
> > > > > > **follow-up on "evaluation"**
> > > > > >
> > > > > > As far as I understand, in your experiment, the attacker and referee still use the same attack.  To call it "attack agnostic", the attacker and the referee need to use different attack manners, then it can follow the definition you mentioned.
> > > > > >
> > > > > > I hope this discussion can be polite and respectful. It's arguably the authors' responsibility to make everything clear. If this discussion continues with such a tone, I do not plan to further contribute my time.
> > > > > >
> > > > > > Also, a reminder that questions in the "Follow up in 'attack agnostic'" are not addressed. If you think I misunderstood your method, the best way might be to report some preliminary results on what I requested (the experiment shouldn't take that long).

---

> > > > > > > ### Author Response · Authors · 2020-11-11
> > > > > > > **There is no attack by the referee**
> > > > > > >
> > > > > > > [We are aware some concerns are not addressed, we will post separately, in order to keep this thread focused]
> > > > > > >
> > > > > > > There is no attack by referee.
> > > > > > >
> > > > > > > For example, please refer to Definition 1 for how to turn classic minimax adversarial robustness into a threat model.
> > > > > > > The evaluation phase, a referee (or nature, which is more commonly used in theoretical studies of crypto) honestly
> > > > > > > evaluates the accuracy of the final model. This is a standard way to make **threat models** amenable to analysis
> > > > > > > (because we can then report the valuation of the game).
> > > > > > >
> > > > > > > Put simply, what happens in our experiments are simply:
> > > > > > > 1. The attacker can perturb arbitrarily (subject attack type) on the natural test set $V$, producing $\widetilde{V}$.
> > > > > > > 2. The defender receives $U=\widetilde{V}|_X$, and produce the adapted model (this is where transductive learning happens) $\widetilde{F}=\Gamma(F, D, U)$
> > > > > > >
> > > > > > > Now, to know how good the defense is, we (i.e., the referee in the threat model) evaluates the accuracy of $\widetilde{F}$ on $\widetilde{V}$.
> > > > > > >
> > > > > > > We are happy to perform experiments you proposed, and we want to be factual and respectful, but the experiment you proposed
> > > > > > > above is not what we mean by transductive defenses.
> > > > > > >
> > > > > > > Finally, we will try our best to revise and clarify the writing. We have been following standard practice in security and
> > > > > > > cryptography, but we acknowledge that there is a gap that makes this paper hard to digest.

---

> > > > > > > > ### Comment · AnonReviewer2 · 2020-11-11
> > > > > > > > **Thanks for clarification.**
> > > > > > > >
> > > > > > > > I see. In that case, you don't have to perform the experiment I requested.

---

### Official Review · AnonReviewer1 · 2020-10-28
**Interesting framework, but requires better motivation and utility**

**Rating:** 3
**Confidence:** 4

**Review:**

This paper studies test-time adversarial robustness through a maximin framework and illustrate non-trivial robustness (under transfer attack) using domain adversarial neural network (DANN) to Linf-norm and unseen adversarial attacks. While I agree that test-time adaptation is an important and practical approach for adversarial robustness, the current version, in my opinion, does not deliver significantly novel insights, nor considering a reasonably practical threat model. My main concerns are detailed as follows.

1. Impractical threat model: At test time, the maximin threat model (Def. 2) assumes the attacker to make move prior to the defender's action, which limits the attacker's ability and weakens the robustness evaluation. More importantly, it may create a false of security/robustness, as pointed out by (Athalye et al. 2018), that the robustness gain may actually come from information obfuscation and thus the results may fail to provide meaningful robustness evaluation. Although the authors mention the maximin threat model is a weaker (attack) model and part of the goal is to find an adaptation method that is "good" in this attack-move-first scenario, I couldn't see the practical utility and contributions form these results. Even in the test-time adaptation setting, the "defender-move-first" setting should be more practical. Better motivation and use cases are needed to justify why the considered setting is important.

2. Due to the assumption of the considered maximin threat model, the experiments are limited to comparing accuracy on transfer attacks, which provide limited understanding of the true robustness of the victim model. Moreover, the baseline models in comparison are too weak and unfair. To have a fair comparison, the authors are suggested to compare robustness on robust models such as TRADES [R1] and adversarially trained models with unlabeled data [R2,R3], so that the baseline models also use unlabeled data. If DANN shows limited robustness against white-box attacks but stronger robustness against transfer attacks, one can only conclude that transfer attack is a weaker threat model, which is a known result. I do not see new insights from the reported results.

[R1] https://arxiv.org/abs/1901.08573

[R2] https://papers.nips.cc/paper/9298-unlabeled-data-improves-adversarial-robustness.pdf

[R3] https://deepmind.com/research/publications/Are-Labels-Required-for-Improving-Adversarial-Robustness

3. In addition to the issue of impractical threat model and lacking motivation, this paper contains too many high-level discussions accompanied with limited or even no empirical evidence. The theorem presented in the paper is a natural use of maximin inequality.  In my opinion, the current version requires significant revision. I suggest the authors carefully motivate the research goals (especially answering why the studied problems and settings are important), consolidate the claims on test-time robustness with convincing evidence, and make a broader connection to other test-time defenses other than DANN.

---

> ### Author Response · Authors · 2020-11-10
> **Transductive adversarial robustness is well motivated**
>
> Thanks for the review. The reviewer challenges our motivation for studying transductive adversarial robustness, which we want to
> rebut right away because this is critical to the understanding.
>
> 1. The threat model we studied in the paper is a particular instantiation of "transductive adversarial robustness", which has recently
> received attention in theory. For example, a novel recent neurips spotlight paper by Goldwasser, Kalai, Kalai and Montasser (https://arxiv.org/abs/2007.05145) has studied very similar setting to ours. In that paper, it has explicitly proved for bounded-VC
> dimension concept class, transductive learning offers important robustness guarantees. Roughly speaking, their formulation is
> more amenable to theoretical study, while ours are more practically relevant for deep learning.
>
> 2. Regarding the "gradient obfuscation" comment, we have explicitly mentioned the differences with "online defenses" and "gradient obfuscation". See pp. 3, paragraph "Test-time defenses and BPDA." Basically, the previous "online defenses" apply to
> a setting where we want to "sanitize" $x$ to $x'$ and then send to a pretrained model $F$. We investigate a much broader setting
> where we can modify model (sometimes completely) to predict $x$.
>
> 3. The reviewer mentioned that "the threat model is impractical". We disagree: Many practical machine learning applications run in a batch mode, where one has received a batch $U$ of test examples (produced by the adversary potentially),
> and we want to be correct on those $U$. In those cases, transductive adversarial robustness is actually very appealing (we want to
> refer the reviewer to Reviewer 3 and 4, who have explicitly commented on this novelty). We have explicitly mentioned this in the introduction, quote: *"First, this question is of practical interest: Many practical ML pipelines run in a batch mode, where they first collect a set of unlabelled data points, and then send them to a model (e.g. Nado et al. (2020)). In such cases,
> data in the batch may have been adversarially perturbed, and it is a natural question whether we can
> leverage the large batch size and test-time adaptation to enhance adversarial robustness"*
>
> 4. We are very confused about one comment, quote *"Even in the test-time adaptation setting, the "defender-move-first" setting should be more practical. "*. What does that even mean? The practical scenario we considered is that in many actual ML pipelines, one received an unlabeled data set $U$ to classify, which could be corrupted (but must happen before we receive it), and we want to use transductive learning to leverage $U$ to get correct prediction. In such cases, defender moves **after** the attacker moves when test-time adaptation happens.

---

> > ### Comment · AnonReviewer1 · 2020-11-10
> > **More clarification on "defender-move-first-setting"**
> >
> > I appreciate the authors' prompt feedback. Let me further elaborate on why I think the considered threat model is not practical (which also echoes other reviewers' comments).
> >
> > As mentioned in the BPDA paper "Obfuscated Gradients Give a False Sense of Security: Circumventing Defenses to Adversarial Examples" and the more recent paper "On Adaptive Attacks to Adversarial Example Defenses", many defenses are shown to be broken by advanced attacks in the "adaptive" attack setting - meaning the adaptive attack is fully aware of the defense in place and seeks to break it. In this adaptive attack setting, the defender makes the first move, and the attacker tries to break the defense after seeing the defense action, which makes the "defender moves first" setting. On the other hand, as mentioned in the fourth item, this paper considers the "attacker moves first" setting, which assumes the attack is done in the first place (manipulations on U) before the defender tries to fix it. As argued in many papers, including the two particular papers I listed, the "attacker moves first" setting only leads to weak non-adaptive attacks compared to the "defender moves first" setting that considers adaptive attacks, and thus the robustness evaluation is less practical. Without addressing the adaptive attack setting, my concern is that this work may fall into yet another loop hope of being broken by follow-up adaptive attacks, which is not an ideal progress that this community wants to see.

---

> > > ### Author Response · Authors · 2020-11-10
> > > **Thanks for the clarifications, but we disagree with the assessment**
> > >
> > > Now we see more clearly where your comment comes from. But we disagree:
> > >
> > > 1. First, as we pointed out, transductive adversarial robustness has been shown to **provably** work in some surprising settings.
> > > We refer the reviewer to the beautiful recent spotlight paper by Goldwasser et al. which will appear in the incoming  neurips conference: https://arxiv.org/abs/2007.05145. It has shown that for bounded VC concept class, transductive adversarial learning (similar to our
> > > modeling, but designed to make possible theoretical analysis) can even provide robustness for **arbitrary test adv examples**. Our work provides a modeling, similar in spirit, that is amenable to algorithm design in the deep learning setting, and we show that a surprising connection between unsupervised domain adaptation and transductive adversarial robustness. We view these as novel contributions, that corroborates with recent theory advancement.
> > >
> > > 2. We are familiar with BPDA and its arguments. We have explicitly discussed this in a paragraph, quote: *"Test-time defenses and BPDA. Various previous work have investigated test-time defenses where
> > > a pretrained model is fixed and there is a “preprocessing procedure” to preprocess an input before
> > > sending it to the model. Several such defenses were described and attacked in Athalye et al. (2018),
> > > by the BPDA technique (Backward Pass Differentiable Approximation). While syntactically one can
> > > fit test-time defenses into our framework, the test-time adaptation concept discussed in this paper is
> > > significantly different, which usually allows one to completely modify a model. As we will see later
> > > in the paper, we use an unsupervised domain adaptation algorithm, such as DANN, for test-time
> > > adaptation. In these cases, it is unclear how to apply BPDA."*
> > >
> > > 3. To this end, we also understand your comments about "adaptive attacks", in the paper we have explicitly mentioned
> > > the fixed point attacks, which follow from a bilevel optimization formulation where we assume the attackers
> > > are aware of the defense algorithm. We will clarify further in the revision and further comments to Reviewer 2
> > > (who also raised a similar question).
> > >
> > > 4.  Pushing the above points a bit further, they are exactly why the results in this paper are surprising and are worth knowing to the community. To this end, one of our concerns is that Athalye et al. (2018) may have pushed part of the community to a place where only the minimax threat model is regarded as the norm for research, but both our work and Goldwasser's work have given (to us, clear) evidence that this is not the case, and **transductive** adversarial learning may have great potential (e.g., DANN can simultaneously provides robustness for both $\ell_\infty$ and $\ell_2$ attacks, without even leverage the attack type).
> > >
> > > 5. Finally, our conceptual novelty has been echoed by both Reviewer 3 and 4. We hope their comments can clarify further.

---

> > > > ### Comment · AnonReviewer1 · 2020-11-21
> > > > **Follow-up comment**
> > > >
> > > > I thank the authors for reporting new results and providing a revised paper. I still have several concerns based on the updates.
> > > >
> > > > 1. I took a read of the mentioned paper https://arxiv.org/abs/2007.05145. Although they studied a less-popular and restricted threat model setting (transductive robustness, aka transfer attack with a batch of adversarial examples), their major contributions lie in the generality to *arbitrary adversarial test examples*. Does this work also apply? To my understanding, the experiments are shown with norm-bounded attacks rather than being arbitrary.
> > > >
> > > > 2. The adaptive attack results are concerning. Comparing Table 1 and Table 3, with the adaptive attack (J-FPAM), the accuracy in the homogeneous setting is below that of adversarial training (Adv S) in Table 1, which somehow echoes my concern on not testing the transductive setting using strong attacks. It seems that adversarially trained models can better defend the adaptive attacks. Moreover, since Adv S is not the most advanced adversarial robust model, I strongly suggest the authors compare the proposed method to more fair and advanced robust models, such as TRADES, Adv Training + unlabeled data, as I pointed out earlier.

---

> ### Author Response · Authors · 2020-11-21
> **Replies to your [questions](https://openreview.net/forum?id=RkqYJw5TMD7&noteId=10S-78b-ji-)**
>
> [Start a new thread to avoid excessive nesting]
>
> We thank the reviewer for the prompt replies. Belows are our replies:
> 1. Your most important concern, we believe, is about comparing with more state of the art adversarially trained models, such as TRADES. While we understand the motivation, the goal of this work is **not to show that DANN provides better robustness than the SOTA adversarially trained models**. Rather, the goal is to demonstrate that **DANN is a candidate in deep learning, that provides robustness in the maximin model, but not in the minimax model**. In our view, **the comparisons with adversarially trained models should only be interpreted at a qualitative level, and improving transductive robustness belongs to future work**. To this end, J-FPAM follows SOTA bilevel optimization work in deep learning, and we believe that it is representative for adaptive attacks in this setting. We will revise the writing so that these points are made clear (we realized that the presentation of our tables may have caused some confusion about our goal).
>
> 2. Now back to your first question. The reason that Goldwasser et al.'s work applies to arbitrary perturbations critically hinges on the condition of bounded VC dimension. In fact, the intuition behind why their Rejectron algorithm works is that they train an ensemble based on U (the transductive input) and use agreement in the ensemble to filter out bad points. Then, the bounded VC dimension allows them to argue line 2 of their Rejectron algorithm to have nontrivial guarantees.
>
> 3. So no, we don't think that DANN can be applied to arbitrary perturbations, but the fact that DANN (which is not explicit adversarial training) can provide adversarial robustness against norm based attacks (we demonstrated for both $\ell_\infty$ and $\ell_2$ attacks), is of significant interest (and further, to us, either confirming this or refuting this by the community, probably in the future, is very interesting).

---

### Official Review · AnonReviewer3 · 2020-10-29
**Interesting topic and solid analysis**

**Rating:** 7
**Confidence:** 2

**Review:**

The paper explores adversarial robustness in a new setting of test-time adaptation. It shows this new problem of “test-time-adapted adversarial robustness” is strictly weaker than the “traditional adversarial robustness” when assuming the training data is available for the “test-time-adapted adversarial robustness”. The gap between the two problems is demonstrated by the simple DANN solution which has good “test-time-adapted adversarial robustness” but bad “traditional adversarial robustness”. The paper also explores the subcase of “test-time-adapted adversarial robustness” when assuming the training data is not available and provide some initial result.

The paper has clear strong points. It aims to tackle an important problem, the “test time adaptation allowed” extension for the “adversarial robustness”. The paper has a nice global picture and a clear position for this piece of work. I particularly like the way the author approaches the problem. They start from the most abstract and fundamental question, “test-time-adapted adversarial robustness” v.s. “Classic adversarial robustness”, or transductive learning v.s. Inductive learning in the setting of adversarial robustness. To proceed the thinking, they develop a good theoretical framework (the two threat models from definition 1 and definition 2) to formulate the two problems. And then consider a middle setting (definition 3) between the classic minimax and new maximin threat model.  Such frameworks help to develop theoretical understanding like one setting the strictly weaker than another (proposition 1).

The weak points of the paper are mainly about the presentation. The paper currently is very dense. Sometimes I feel the author may assume the reader has certain domain knowledge without explanations. For example, dataset “CIFAR10c-fog” appears very early in the introduction, but it is never clearly explained what it is, what is the main difference between it and CIFAR 10. After reading the paper the only impression I get is they not homogeneous. There also some places in the method, I can not understand,
For the maximin threat model, why the game is maximation over U instead of A (the left side of the equation from proposition 1).
Why there are A0 and A1 in the Adversarial semi-supervised minimax threat model? And why in this game, A0 and A1 are jointly maximizing L(\tile{F},\tilde{V}’)?

More question about experiments:
For FPA attacks, is there any baseline method we can compare the DANN with? Currently, I am not sure how to evaluate the performance of DANN.
In experiment (D), it says, “we also evaluate the accuracy of the adapted DANN models in the minimax threat mode”. But where are the results?

I understand it is pretty hard to squeeze so many contents into limited 8 pages. Personally, I think it is helpful to cut off some content and make the main paper more clear, well organized, and strong.

Unfortunately, I am not an expert in adversarial robustness. I did not check the technique and experiments deeply. My current score assumes no fatal flaws exist in the theory and experiment. My rating will be changed according to other reviewers’ comments and the author’s updates.

---

> ### Author Response · Authors · 2020-11-11
> **Thanks for recognizing our conceptual novelty, and some first clarifications**
>
> We thank the reviewer for the positive feedback, and recognizing our conceptual novelty. Below we want to address some  questions that we think are important to clarify first:
>
> 1. CIFAR10-c data set (https://github.com/hendrycks/robustness). The  CIFAR10-c dataset is an important recent benchmark for evaluating the robustness of learning in the setting of adaptations, under common corruptions and perturbations.  Since its introduction, this dataset has been very commonly used in domain adaptation setting, and in particular, it is used in TTT (Test Time Training, a oblivious test-time adaptation algorithm) to demonstrate that test-time adaptation can achieve robustness.
>
> 2. In our work, we naturally generalize the consideration, and consider whether test-time adaptation can achieve, not only robustness on common corruptions and perturbations, but also on **adversarial perturbations**. We have indeed assumed that
> readers are familiar with these points. We will clarify the writing by adding relevant details.
>
> 3. Why the adversarial game is over $U$? This is because that the adversary can only perturb features, but not labels. Therefore, to
> make this point precise, we write the maximum to be over $U$ (we thank the reviewers to notice these subtleties).
>
> 4. Why there are two different attack algorithms in the semi-supervised adversarial model? This is explained in Section B.1, paragraph **"Why do we need to have two algorithms A0, A1 in Definition 3?"** Very roughly speaking, the first attack is about attacking the semi-supervised learning algorithm, and the second attack is about attacking the final model we learned.
>
> 5. DANN results in the minimax model. We have a sentence *"Not surprisingly, the accuracy becomes very low"* (basically, the
> accuracy drops to random guessing ~10%). We choose to present in this way because this experiment is simple to perform, and
> the results are very much as expected.
>
> 6. We have clarified the point regarding the baselines for FPA in the thread "Revision #1: Clarifying adaptive fixed point attacks".

---

### Author Response · Authors · 2020-11-11
**Revision #1: Clarifying adaptive fixed point attacks**

We note that all reviewers have raised questions about our fixed point attacks, therefore we have prioritized clarifying it
(we are working hard on revising concerns on experiments). In the revision, Section D (MORE ON FIXED POINT ATTACKS),
you can find a new section about details of deriving the adaptive fixed point attacks. We list some important points that may help reviewers digest:

1. The derivation of FPA follows a **bilevel optimization formulation** of maximin game (please see equation (1) in Section D).
The bilevel optimization nature of test-adaptation based defense differs significantly from previous test-time defenses (as those
discussed in the BPDA work), where none of those defenses amount to bilevel optimization (because they are about sanitizing
input with a fixed pretrained model, instead of "freely" optimizing for models). This is related to a concern raised by **Reviewer 1**
below in the discussion.

2. One should note that bilevel optimization with deep nets is very challenging (but this is good news to the defender).To arrive at FPA, we actually adapted an algorithm proposed by Jonathan Lorraine and David Duvenaud (Specifically, see  Algorithm 2 (optimization of hypernetwork, then hyperparameters) in "Stochastic hyperparameter optimization through hypernetworks"). In fact, their algorithm is essentially FPA  with $k=1$, and we extend the algorithm to $k>1$. We apologize that we have not clarified the root of FPA (partially, as Reviewer 3 has pointed out, this paper has been very dense, that we have been assuming some background knowledge without describing all details).

3. **Reviewer 4** mentioned, quote: *"I would imagine a strong adaptive attack could be built by generating adversarial inputs with the objective of minimizing the loss of DANN's domain label classifier."* We have indeed tried something along this line by directly attacking the DANN objective, but it works less effectively than the FPA and transfer attacks we considered. Nevertheless, we don't know whether that matches exactly what's in your mind: Can you elaborate the exact objective in your mind? We can experiment that immediately.

4. **Reviewer 4** also mentioned, quote: *"In particular its effectiveness in the homogeneous setting for CIFAR-10 reduces with the number of iterations k. This does not seem to be a reliable way of evaluating DANN's maximin robustness"* Since there is no theory guarantees, it is hard to say, but FPA is principled and we believe that this phenomenon is because DANN makes the convergence of solving the bilevel optimization really hard. This is indeed good news for the defender.

5. Finally, **Reviewer 3** mentioned, quote: *"For FPA attacks, is there any baseline method we can compare the DANN with?"* That's good question. We are using FPA attacks to, hopefully, reduce the maximin robustness provided by  DANN, as much as possible, so as to show that maybe DANN can indeed be attacked (so the baseline here is really the SOTA robustness numbers achieved in the minimax threat model, where we hope in maximin threat model life can be easier). Of course, FPA is so far not effective as to break DANN. On the other hand, if your question is whether there is a baseline to compare with DANN on maximin robustness, then unfortunately there is no: FPA attacks,  the conception of maximin threat model, and DANN as a defense, are all novel as far as we are aware of.

---

> ### Comment · AnonReviewer4 · 2020-11-11
> **Reliable robustness evaluation**
>
> I acknowledge that bilevel optimization with deep nets is challenging. However, this is a mixed blessing: the defense (like many other defenses) makes designing effective attacks more difficult. This in turn makes evaluating the actual robustness challenging (and this actual robustness is what we really care about).  So that "DANN makes the convergence of solving the bilevel optimization really hard" is potentially "good news for the defender" but "bad news" for the paper: the only thing we know for sure is that the particular attacks tested in this paper fail. But we do not know at all if DANN actually improves robustness or if "next week" someone tries an alternative attack that is vastly more effective than FPA and removes all presumed robustness gains by DANN.
>
> I would encourage the authors to consult "On Evaluating Adversarial Robustness" by Carlini et al. (https://arxiv.org/abs/1902.06705). It contains many practical ways of improving the robustness evaluation of a model. For instance, gradient-free attacks should be tried, such as e.g. Square Attack (https://arxiv.org/abs/1912.00049). (I do not imply that only adding Square Attack is sufficient, it would be just one additional evidence)

---

> > ### Author Response · Authors · 2020-11-12
> > **Thanks for the pointer, and some high-level thoughts on reliability (also related to Reviewer 1)**
> >
> > [We are still working on revising the experiments, but we feel that it is useful to share some of our high level thoughts with the reviewer on this important topic]
> >
> > 1. Thanks for the pointer. We will use the playbook as a guide and check our experiments. We also acknowledge the point that bilevel optimization makes the life of evaluation difficult.
> > 2. On the other hand, while we agree completely to check the reliability of our experiments according to the playbook, we want to push a bit on the point of **"What happens if DANN is broken after all?"** (this is something **Reviewer 1** also mentioned), from a different angle: Transductive adversarial learning is a new territory (if Goldwasser et al.'s work defines it in the case of bounded VC situation, this work defines it in the case of deep learning). This consideration of transductive learning has led to new discoveries that, as far as we know, none of the vast literature of online defenses has observed: **If you free yourself from considering sanitizing inputs, and instead turn to revising models, a simple application of maximin inequality tells that transductive adv learning is theoretically easier**, and further **optimal attack is a bilevel optimization, theoretically** (unlike online defenses which are too rigid). On the more practical side, the discovery of the connection with Unsupervised Domain Adaptation, as well as the brittleness of oblivious adaptation algorithms under transfer attacks, are also results that are first of their kind.
> > 3. So, to us, the proposal of FPA (we know it is simple, but it is principled as we generalize Lorraine&Duvenaud) is a first attempt of an algorithmic framework to understand attacking *truly* transductive defenses. To this end, blatantly, we do not actually worry too much that DANN may eventually get broken. In fact, we believe that both outcomes (either it gets broken eventually, or it stays robust) are good to the community for scientific purposes (so to **Reviewer 1**, we also do not worry about a new iteration of cat-mouse game, because we think it is useful to the community to understand more deeply transductive adv robustness, especially in the setting of deep learning.).
> > 4. To push (2) and (3) a bit further, we also do not worry about the computational resources being consumed by DANN. The use of DANN is an illustration of the power of transductive learning, and improving the resource consumption, running time, etc, are natural future directions. To use a remote, possibly subjective, analogy, it is like in studying computational complexity of algorithms, we want to first find a polynomial-time algorithm, before we make it, say, linear time.
> > 5. As a related remark, you mentioned in one of your comments, quote *"I would imagine a strong adaptive attack ... with the objective of **minimizing the loss of DANN's domain label classifier**. "* We in fact have experimented
> > directly attacking the DANN objective in that fashion (although we do not know whether that matches exactly what's in your mind).  We will include that result into the draft. To this end, we feel in this new deep learning setting of transductive adv robustness, all we can do is the best we can, and it might be too much to ask for a "perfect defense" in a paper where we just manage to define such defenses.
> > 6. As a final related remark, we are also aware of the phenomenon that the FPA performance is not stable with large $k$. One explanation we have is that (we will check more and report) since DANN always applies after $U$ is produced, it makes the convergence of finding fixed points quite difficult. So we are not sure whether we want to call this a strong evidence that the attack is too weak. Nevertheless, we understand where your point came from.

---

### Public Comment · ~Chia-Hung_Yuan1 · 2020-11-13
**Similar idea to another paper, Adversarial Robustness via Runtime Masking and Cleansing**

This paper points in an interesting direction, but I found that a similar idea about test-time adaptation and robustness has already been proposed in [Adversarial Robustness via Runtime Masking and Cleansing](https://proceedings.icml.cc/static/paper_files/icml/2020/377-Paper.pdf) at ICML 2020.

This paper used [TTT](https://proceedings.icml.cc/static/paper_files/icml/2020/4786-Paper.pdf) as a baseline model might be inappropriate, since it was not designed for adversarial robustness. I think it is better to compare your model with the abovementioned work.

---

> ### Author Response · Authors · 2020-11-13
> **Thanks for the pointer, we will cite, but they are indeed very different work**
>
> Thanks Chia-Hung for the pointer. We will cite and discuss the paper in the related work.
>
> However,  these two work are very different (especially the detailed threat model and what we want to experiment with, see below):
>
> 1. To start with, the positioning of our paper is *not* any particular defense, but to formalize **a framework for the transductive adversarial learning in the deep learning setting**. This framework includes several specializations that cover special kinds of defenses. For example: (1) **homogeneity** (source domain = target domain), (2) **data-obliviousness** (at test time, we do not need training data), (3) **attack-agnostic** (the test-time adaptation is agnostic to the attack type), and (4) **online** (see Definition 4 in Section B.2).  Within this framework, we examined two special modelings to demonstrate the potential of transductive adversarial learning (we will elaborate this below).
>
> 2. With (1) in mind, we are **not** using TTT as a baseline.  According to our framework, TTT falls into data-oblivious defenses, and we are just using TTT to demonstrate that some recent data-oblivious adaptation proposals are even brittle to simple transfer attacks (as we understand your work, it is not data-oblivious, because it still needs to access $\mathbb{D}$, the training data set). Since data-obliviousness is a desirable property, we want to use this to trigger interests of the community to study data-oblivious adaptations that are adversarially robust. That is our purpose there.
>
> 3. Now, the modeling mentioned in your paper, is indeed a specialization of our framework, which is **homogeneous** (your source and target domain are equal), **online** (because you need to handle these adversarially perturbed examples one by one), but is **not** data-oblivious, and also is **not** attack-agnostic.
>
> 4. The modeling of your defense, though is within our framework (which we will discuss, thanks), is not our focus of experiments. We consider a setting that target domain can be **inhomogeneous**, but the test data comes in **batch**, and most importantly, the defense is **attack-agnostic** (our defense does not use adversarial training, and in fact is agnostic to norm-based attacks, but your defense is not because it still requires specific adversarial training to get $\mathbb{D}'$). The reason we consider this setting is that it shows surprising power of transductive learning, and it can provides  robustness even there.
>
> We hope that this clarifies for you.

---

### Author Response · Authors · 2020-11-18
**Revision #2: Adaptive attacks and clarifying positioning**

We thank the reviewers for taking your time to discuss with us. Based on your feedback, we have prepared a second revision to clarify our positioning of this work, as well as enhancing our arguments on adaptive attacks. A summary of the revisions are as follows:

1. **Positioning and significance of the current work**. As a summary of our discussion below, we stress that the positioning of this work is to formulate **transductive adversarial deep learning**. The study of adversarial robustness lacks precise formulation of threat models (unlike more traditional fields in security and cryptography). Not only our work fills in this gap, but also it has led to **novel connections to Unsupervised Domain Adaptation as defense mechanisms**, for which none of the previous test-time defenses has considered. To this end, we do not view this work as incremental to previous work, or a summary of previous online-defenses, but rather, it significantly broadens the scope of "online defenses".  We noticed that Section 2.2 in ["On Evaluating Adversarial Robustness"](https://arxiv.org/abs/1902.06705) has also emphasized the importance of threat models, which corroborates our argument here.

2. **DANN is a *candidate* adaptation that demonstrates a separation.** As we mentioned below (point 3 of [this discussion](https://openreview.net/forum?id=RkqYJw5TMD7&noteId=0mRXYdzBy5Y)), we don't worry that DANN gets broken by some future adaptive attacks. In fact, it has always been our purpose in this paper that we want to say that DANN is a **candidate** algorithm that demonstrates a separation between maximin and minimax threat models (much like in security and cryptography studies, we have candidate schemes). We thus have revised the writing to clarify this message everywhere. In particular, in Section 4.1, we have explicitly called DANN a **candidate** adaption. To this end, we stress again that breaking DANN itself could also be a very interesting result, because, as far as we know, no one has deeply studied advesarially attacking DANN, except perhaps for this work.

3. **New adaptive attacks (our hypothesis that DANN is a candidate separation still holds).** Nevertheless, we agree with the reviewers that even if the goal is to show DANN is a candidate, adaptive attacks need to be further investigated. To this end, we have identified a relaxation of the bilevel optimization so that we can iteratively attack DANN objective, much like for the traditional minimax setting. Please refer to Appendix D, Algorithm 4 (**Joint Fixed Point Alternating Method**). Our most effective so far (see Table 3), on CIFAR10 related tasks, has brought the robustness to about 40% in the homogeneous case, and to 30% in the inhomogeneous case. Nevertheless, this is still nontrivial robustness compared to 0% robustness of DANN in the minimax threat model, and still support **our hypothesis that DANN is a candidate separation** (note that in the inhomogeneous case, DANN still performs better than explicit adversarial training, which, as we argued below, is natural and interesting).

4. **Clarification of oblivious adaptations.** We have also clarified the last section of the paper, by changing its title to "ROBUSTNESS OF OBLIVIOUS ADAPTATION ..." We clarified that oblivious adaptations are even vulnerable under transfer attacks, which is in sharp contrast with our results on DANN. We have also emphasized the future work of identifying more effective attacks against DANN (it was there, but now we emphasized more about it).

5. **More ongoing work**. We still have ongoing work to further revise the paper, and we will update once we feel they are ready. For the experiment on batch size that **Reviewer 2** raised, we want to remark that this particular experiment is simple and much as we expected that DANN works worse for smaller batch size (for example, if batch only has 1 point, it is unlikely to work), so we are not exactly sure that more repeated experiments would contribute to new interesting findings. Nevertheless, we have put it on our task list, and will try the best we can.

6. **Release of anonymized code base**. Finally, we will soon release an anonymized repo of our code, and include experiment details into the draft.

---

> ### Comment · AnonReviewer4 · 2020-11-18
> **Early feedback on the revision**
>
> I acknowledge that the authors invest a lot of additional work during the response period, this is highly appreciated. From my point of view, the main question however remains:
> _Is there any reliable evidence that DANN actually separates robustness in the test-time maximin threat model compared to the minimax threat model?_
> The authors themselves state that only if this is demonstrated, the experimental goal is achieved (top of page 7). In this respect, it is in my view not sufficient to state DANN being a "candidate" adaption that later work might be able to break. Because this positioning of DANN leaves it unclear if maximin and minimax are actually separated or if the current gap is only due to "weak" attacks in maximin (I acknowledge there is progress on stronger attacks but the principle issue remains). In my point of view, it would be more helpful to have some results on toy data which however demonstrate this separation in a "bullet-proof" way. This would establish that the test-time maximin threat model is actually fundamentally more difficult for the attacker (which we don't know for sure at the current state).
> Work on scaling defenses and attacks to actual datasets could then be a next step.

---

### Author Response · Authors · 2020-11-21
**Revision #3: Provable separation, a toy example for DANN, and request for R4's clarifications**

We thank R4 for the prompt feedback about our second revision. The questions raised indeed hit at the core of this paper: **What really constitute clear and convincing evidence for showing the separation between the test-time maximin threat model and the (semi-supervised) minimax model? And further what about this for DANN in particular?**

We give below several responses to these two questions. Most importantly, we provided **a provable separation (in the new Section 4.2) of the test-time maximin and minimax threat models**, which thus resolves the first question above. All these points have been collected into the revised draft. Below we give a summary of important points:

1. **Valuation of the game**. In terms of the valuation of game (along the lines of our Proposition 1 and 2), test-time maximin model is strictly weaker than minimax if the concept class has unbounded VC dimension (let's restrict to binary classification tasks). The proof is simple: If the concept class has unbounded VC dimension, then given $\widetilde{V}$, a good model $\widetilde{F}$ exists to fit data in $\widetilde{V}$, while fitting perfectly on the training data $D$. Therefore the **valuation** of the maximin game in this case is **always $0$.** Note that this argument critically relies on transductive learning for predictions specifically on $U$. This is now recorded in Proposition 2 in Section 4.1 of the revised draft.

2. **Is there a setting where we have a separation between maximin and minimax with a real adaptation algorithm?** Clearly, however, argument in (1) has not really addressed the core question because even though $\widetilde{F}$ exists, it does not mean that there is an adaptation algorithm which can identify such $\widetilde{F}$ with only unlabeled data $U$. To this end, as a first point, we note that the work of [Goldwasser et al.](https://arxiv.org/abs/2007.05145) has provided **strong evidence that this is the case (though strictly speaking, they did not prove lower bound for minimax model)**, even though their modeling is a bit different as they allow redaction (and some other restrictions that are too stringent for deep learning). Specifically, they have shown that in **a bounded VC-dimension scenario**, there is a transductive algorithm (with redaction), namely Rejectron algorithm (see section 4.2 of their paper), that can handle *arbitrary adversarial perturbation*. Such guarantees seem unlikely to hold in the minimax threat model.

3. In the now new Section 4.2 "Provable Separation in a Gaussian Data Model" (see Theorem 1 and its proof in the Appendix F), **we provide a provable separation, and thus resolves the question in (2)**. The algorithm is essentially a form of transductive SVM, and leverages $U$ to build two classifiers, and finally uses part of the training set to validate the errors, and classify $U$ using the winner classifier. This natural transductive algorithms leads to a provable separation of the maximin and minimax threat models. In fact, the separation holds **even when $U$ contains only one point, indicating the power of transductive learning**.

4. Finally, how about all these for DANN? It is hard to prove DANN's performance theoretically (that is a generally hard topic). Specifically, to demonstrate a separation with DANN, we believe that one must have complex model boundary (which implies complex data). Nevertheless, if the question is whether there is a toy data set on which ** we can demonstrate DANN actually works** (well, since we know that DANN in the deep learning setting fails in the minimax threat model, identifying a toy dataset on which DANN works is of sufficient interest to us), then we can do something below.

5. To this end, we note the original paper of DANN (https://jmlr.org/papers/volume17/15-239/15-239.pdf), Section 5.1.1, Figure 2, already provided such an example (and it can be interpreted exactly as transductive learning): There, the adversary rotates the labeled data by $35^{\circ}$, and send them as $U$ for DANN to fit. DANN correctly learns the boundary (see part (b)), while a natural training fails on boundary points (inductive learning, see part (a)). This example can be easily generalized to a case with bounded degree rotations.

6. **Our question for R4** is: **Is our interpretation of your comments correct?** Clearly, it is challenging to exhibit an ideal separation for DANN, but we believe that question itself is an important future work that can trigger much interest to the community (again, as far as we know, no work has considered even adversarially attacking DANN).

---

### Author Response · Authors · 2020-11-22
**Revision #4: Minor edits and clarifications**

We have done a pass of minor edits (hopefully the last revision during the rebuttal) to clarify the writing. Some important points are summarized below (respective reviewers are highlighted):

1. **[R4]** We have revised the introduction, and also the beginning of Section 4.2 to clarify the significance of the provable separation. Note that the separation holds in a natural and simple Gaussian data model (which is advantageous in terms of theory results), and it holds even when the adaptation algorithm only receives **one** unlabeled data point. This gives indication of the power of transductive learning.

2. **[R3]** We have revised the introduction and removed reference to CIFAR10c. We added a sentence in the **Datasets** paragraph to introduce CIFAR10c, quote: *"CIFAR10c is a recent benchmark for evaluating neural network robustness against common corruptions and perturbations."*

3. **[All reviewers]** We have further clarified BPDA and test-time defenses. See the paragraph **Test-time defenses and BPDA**. We added, quote: *"While syntactically one can
fit test-time defenses into our framework, they only form some very special cases of our framework, which reuses a fixed pretrained model and focuses on input sanitization. As we will show later in the paper, for both of our provable separation and deep learning results, the adaptation algorithm critically relies on training new models (rather than sanitizing inputs); and theoretically attacking these adaptations becomes a bilevel optimization."*

4. **[All reviewers]** We have added a sentence about studying attacking DANN further, in the last section, quote: *"To this end, either confirming (theoretically, for example) or refuting the maximin robustness of DANN is intriguing."*.

5. **[All reviewers]** We have included experiment details in Section E.4. Our experiments are fairly standard so reproducibility should not be an issue.

6. **[R2]** Due to lack of time and space, and that the results should very much match the intuition (DANN will need large enough batch size in order to match distribution), we decide not to follow further on the batch size experiment in the rebuttal. However, if this paper has a chance to appear in ICLR, we will include more results in the appendix when preparing the final version (which to us is a minor point).

---

### Author Response · Authors · 2020-11-24
**Revision #5: Code submitted and some remarks**

We thank the reviewers for the reviews. Please refer to the supplementary material for the code if needed.

Two nits from R4 that we found that we did not address when we read the reviews again:

1. *"Threat models could mention that white-box knowledge of the attacker is assumed"*: The attacker algorithms have encoded the white-box knowledge, since the attacker algorithm is ${\cal A}(\Gamma, F, D, V )$ (note that the attacker is aware of the target labels).

2. *"it is not really clear why test inputs are assumed to be labeled in the threat models"*: The attacker knows the target labels, but the defender only sees the unlabeled data. Labels are necessary for the evaluation by the referee (which does not belong to either the attacker, or the defender). For the details please check again the attacker and defender steps at the test time.

As a final remark, we believe that the current work provides a novel and principled foundation for transductive adversarial robustness, and we have provided solid evidence at various levels (conceptual, definitional, theoretical, and empirical).
As a simple example, it **enables** one to draw new connections between unsupervised domain adaptation and adversarial robustness, which has not been considered before, to the best of our knowledge. To us, this alone could fuel many intriguing future works (see the Conjecture we put on pp. 9).

---

### Decision · Program_Chairs · 2021-01-07
**Final Decision**

**Decision:**

Reject

**Comment:**

This paper studies test-time adaptation in the context of adversarial robustness. The key idea is to use a maximin framework, which illustrates non-trivial robustness (under transfer attack) using domain adversarial neural network (DANN) to Linf-norm and unseen adversarial attacks. The approach is sound, well grounded, and quite logical. Results demonstrate the effectiveness.

However, there exists some limitations, for example, 1) The adaptive attack results are concerning. Comparing Table 1 and Table 3, with the adaptive attack (J-FPAM), the accuracy in the homogeneous setting is below that of adversarial training (Adv S) in Table 1, which somehow echoes my concern on not testing the transductive setting using strong attacks. It seems that adversarially trained models can better defend the adaptive attacks. 2) The paper says "This threat model enables us to study whether a large test set can benefit a defender for adversarial robustness", yet there is no any experiments in the main paper that correspond to this discussion. The appendix seems lacking this discussion either. 3) Due to the page limit, a lot of details have been moved into the appendix, making the paper difficult to read.

In the end, I think that this paper may not be ready for publication at ICLR, but the next version must be a strong paper if above limitations can be well addressed.